# GTPBD: A Fine-Grained Global Terraced Parcel and Boundary Dataset

**Zhiwei Zhang**[1], **Zi Ye**[1], **Yibin Wen**[1], **Shuai Yuan**[2],
**Haohuan Fu**[3,4], **Jianxi Huang**[5,6], **Juepeng Zheng**[1,4,*]

[1]Sun Yat-Sen University   [2]The University of Hong Kong   [3]Tsinghua University
[4]National Supercomputing Center in Shenzhen
[5]Southwest Jiaotong University   [6]China Agricultural University

## Abstract

Agricultural parcels serve as basic units for conducting agricultural practices and applications, which is vital for land ownership registration, food security assessment, soil erosion monitoring, etc. However, existing agriculture parcel extraction studies only focus on mid-resolution mapping or regular plain farmlands while lacking representation of complex terraced terrains due to the demands of precision agriculture. In this paper, we introduce a more fine-grained terraced parcel dataset named **GTPBD** (**G**lobal **T**erraced **P**arcel and **B**oundary **D**ataset), which is the first fine-grained dataset covering major worldwide terraced regions with more than 200,000 complex terraced parcels with manually annotation. GTPBD comprises 47,537 high-resolution images with three-level labels, including pixel-level boundary labels, mask labels, and parcel labels. It covers seven major geographic zones in China and transcontinental climatic regions around the world. Compared to the existing datasets, the GTPBD dataset brings considerable challenges due to the: (1) terrain diversity; (2) complex and irregular parcel objects; and (3) multiple domain styles. Our proposed GTPBD dataset is suitable for four different tasks, including semantic segmentation, edge detection, terraced parcel extraction and unsupervised domain adaptation (UDA) tasks. Accordingly, we benchmark the GTPBD dataset on eight semantic segmentation methods, four edge extraction methods, three parcel extraction methods and five UDA methods, along with a multi-dimensional evaluation framework integrating pixel-level and object-level metrics. GTPBD fills a critical gap in terraced remote sensing research, providing a basic infrastructure for fine-grained agricultural terrain analysis and cross-scenario knowledge transfer. The code and data are available at `https://github.com/Z-ZW-WXQ/GTPBD/`.

## 1 Introduction

In ancient China, terraced fields were given the reputation of "Dragon Spines" ("龙脊" in Chinese), which is not only a poetic description of their winding and magnificent forms, but also a high praise in agricultural civilization. Nowadays, approximately 120 million acres of terraced fields worldwide still support more than 500 million mountainous populations [36], reducing soil erosion by more than 23.7 billion tons annually, which indicates significant ecological and economic values [12]. Therefore, a better and fine-grained understanding of terraced area is needed for a sustainable agricultural ecosystem, which is vital for land ownership registration, food security assessment, etc [30, 2].

With the development of artificial intelligence techniques, terrace mapping have progressed significantly, from pixel-based and object-oriented methods to machine learning and deep learning-based

---

*Corresponding author. Email: `zhengjp8@mail.sysu.edu.cn`

39th Conference on Neural Information Processing Systems (NeurIPS 2025) Track on Datasets and Benchmarks.

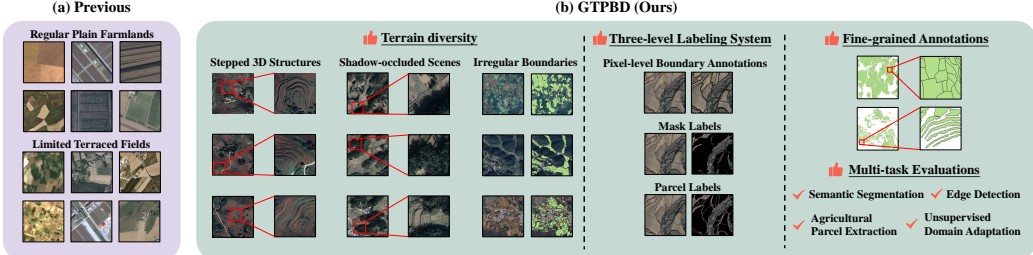

Figure 1: Comparisons between previous dataset that focus on regular plain farmlands with limited terraced fields, and our GTPBD that offers more challenging and comprehensive evaluations.

methods from remote sensing images. Existing terrace mapping only focus on mid-resolution mapping [4, 21, 11] using Sentinel-1/2, Landsat, MODIS, etc. Meanwhile, some researchers have collected public large-scale agricultural parcels and boundaries datasets from high-resolution remote sensing images. To improve the agricultural parcel extraction, advanced deep learning methods have been used in parcel-level extraction and vectorization [44, 26, 22, 19].

Despite the impressive capabilities demonstrated by deep learning for agricultural parcel extraction, fine-grained terraced parcel and boundary benchmarks for agricultural remote sensing remain scarce due to they are **extremely complex and unexpectedly irregular**. Existing agricultural parcel dataset primarily focus on scenarios of regular plain farmlands (See Fig. 1 (a)). Datasets designed for agricultural parcel dataset from high-resolution remote sensing images exhibit notable limitations, primarily in terms of **terrain diversity**, **annotation difficulty** and **multiple tasks**, as shown in Fig. 1.

To begin with, current parcel dataset suffer from regular plain farmlands and limited terraced fields [26, 10], limiting their ability to insufficient terrain diversity (See Fig. 1 (a)). To achieve a more precise agricultural mapping and yield estimation for global terraced field, it is imperative to develop a more fine-grained terraced dataset covering terrain diversity that contains stepped 3D structures, shadow-occluded scenes and irregular boundaries that combine three-dimensional topological structures (See Fig. 1 (b)). In addition, most of existing related datasets only provide binary classification mask labels [51, 10], failing to explicitly distinguish between two topological relationships of adjacent terraced ridges: shared edges (where adjacent parcels share the same boundary) and non-shared edges (where adjacent parcels have separate boundaries). It is highly demanded to collect a refined terraced parcel dataset with fine-grained boundaries. Furthermore, existing agricultural parcel dataset often rely on oversimplified task design such as binary classification or semantic segmentation [28, 39], and have not discussed the transferability of agricultural parcel extraction. In order to improve and benchmark their generalizability, appropriate dataset that includes multiple domain styles is required.

In response to the aforementioned issues, this paper proposes a more challenging terraced parcel dataset named **GTPBD** (**G**lobal **T**erraced **P**arcel and **B**oundary **D**ataset), which is the first fine-grained dataset covering major worldwide terraced regions with more than 200,000 complex terraced parcels with manually annotation.

In summary, the contributions of this work can be concluded as:

• We construct GTPBD which is the first global terraced dataset on parcel level, containing 47,537 images and collectively covering 885 $km^2$ of terraced farmland across varied terrains, with mountainous terraces accounting for over 80% of the dataset. The dataset covers seven major geographic zones in China and transcontinental climatic regions around the world.

• We propose GTPBD, which contains three-level labels for each pixel, including boundary labels, mask labels, and parcel labels. GTPBD is suitable for four different tasks, including semantic segmentation, edge detection, terraced parcel extraction and unsupervised domain adaptation (UDA) tasks with three different domains and six transfer tasks.

• We conduct a comprehensive evaluations of 8 semantic segmentation methods, 4 edge detection methods, 3 parcel extraction methods and 5 UDA methods, long with a multi-dimensional evaluation framework integrating pixel-level and object-level metrics. Experimental results reveal the limitations of current deep learning methods in terraced parcel extraction and its generalization capacity.

Table 1: Comparison between GTPBD and the main agricultural field datasets.

| Dataset | Source | Images | Area (km²) | Resolution (m) | Classes | Global Coverage | SS | Task APE | Task ED | UDA |
|---|---|---|---|---|---|---|---|---|---|---|
| FHAPD [51] | GaoFen-1/2 | 68,982 | <1000 | 1-2 | 2 | | ✓ | ✓ | ✓ | |
| GTM [21] | Sentinel-2 | 108,300 | 853,161 | 10 | 2 | ✓ | ✓ | | | |
| FTW [18] | Sentinel-2 | 70,462 | 166,293 | 10 | 2,3 | ✓ | ✓ | ✓ | ✓ | |
| AI4Boundaries (Sentinel-2) [10] | Sentinel-2 | 7,831 | 51,321 | 10 | 2 | | ✓ | ✓ | ✓ | |
| AI4Boundaries (Ortho) [10] | Aerial | 7,598 | 1,992 | 1 | 2 | | ✓ | ✓ | ✓ | |
| PASTIS [13] | Sentinel-2 | 2,433 | 3,986 | 10 | 19 | | ✓ | ✓ | ✓ | |
| CropHarvest [35] | Sentinel-1/2 | 95,186 | >5,000,000 | 10-60 | 348 | ✓ | ✓ | ✓ | ✓ | |
| GFSAD30 [32] | Landsat-8 | 64,800 | 18,740,000 | 30 | 2 | ✓ | ✓ | | | |
| GloCAB [14] | Sentinel-2 | 190,832 | N/A | 10 | 2 | | ✓ | ✓ | ✓ | |
| AI4SmallFarms [27] | Sentinel-2 | 62 | 1,550 | 10 | 2 | | ✓ | ✓ | ✓ | |
| India Smallholder Boundaries [39] | Airbus SPOT-6/7 & PlanetScope | 10,000 | 30-50 | 1.5-4.8 | 2 | | ✓ | ✓ | ✓ | |
| **GTPBD (Ours)** | **GF-2 & Google Earth** | **47,537** | **885** | **0.5–0.7** | **3** | ✓ | ✓ | ✓ | ✓ | ✓ |

• **SS**: Semantic Segmentation; **APE**: Agricultural Parcel Extraction; **ED**: Edge Detection; **UDA**: Unsupervised Domain Adaptation.

## 2 Related works

### 2.1 Agricultural parcel and boundary datasets

In recent years, considerable progress has been made in developing agricultural parcel and boundary datasets (see Table 1). Early datasets such as GFSAD30 [32] provide global coverage but at a coarse resolution of 30 m, making them unsuitable for fine-grained parcel delineation. More recent efforts like PASTIS [13] and FTW [18], built on Sentinel-2 imagery (10 m resolution), improved spatial granularity but still mainly represent regular, planar agricultural fields. Higher-resolution datasets such as FHAPD [51] and AI4Boundaries(Ortho) [10] offer imagery at 1 m resolution, enhancing spatial detail. However, these remain limited to structured flat farmland, with little focus on irregular terraced landscapes. Importantly, none of the existing datasets support unsupervised domain adaptation (UDA), despite its necessity for cross-region generalization.

To overcome these limitations, we introduce a more challenging terraced parcel dataset named **GTPBD**, specifically designed to address the gaps in existing agricultural datasets. GTPBD is distinguished by its global scope, encompassing diverse regions spanning tropical, subtropical, and temperate climatic zones. Furthermore, GTPBD supports multiple tasks, including semantic segmentation, agricultural parcel extraction, unsupervised domain adaptation, and edge detection, offering a unified benchmark for advancing the analysis of complex terraced agricultural landscapes.

### 2.2 Remote sensing tasks

**Semantic Segmentation (SS)** assigns a class label to every pixel in imagery. Early encoder–decoder CNNs such as U-Net [29], DeepLabV3 [6], and PSPNet [52], recover spatial detail via skip connections, atrous convolutions, and pyramid pooling. More recently, transformer-based frameworks including SegFormer [41] and Mask2Former [8], leverage self-attention to capture long-range dependencies and improve robustness to local appearance variations [8, 41, 5, 54]. Despite their advances, these models have been benchmarked almost on coarse-grained land-cover datasets [51, 1, 34], which focus on urban scenes or flat fields. To fill this gap, we evaluate all of the above architectures on GTPBD, which includes fine-grained annotations, to deliver the comprehensive semantic segmentation results for complex terraced agricultural landscapes.

**Agricultural Parcel Extraction (APE)** delineates farmland units for precision agriculture [48, 37]. Most CNN-based methods treated APE as single-task segmentation, using encoder–decoder backbones to predict parcel masks but often yielding unclosed or jagged boundaries in heterogeneous fields [25]. To address this, edge-aware multi-task architectures have emerged: SEANet [19] jointly predicts masks, distances, and edges to enforce closed parcels, REAUNet [22] integrates Sobel filters and attention for multi-scale edge enhancement, HBGNet [51] extracts low-level boundary features and high-level parcel semantics in parallel with a dual-branch framework. Despite these models perform well on planar benchmarks like FHAPD [51] and AI4Boundaries [10], their robustness to terraced ridge geometries is untested. We therefore benchmark them on GTPBD to provide systematic evaluation of APE methods in complex terraced agricultural landscapes.

**Edge Detection (ED)** precisely localizes pixel-level boundaries and facilitates high-fidelity terrace morphology analysis and erosion monitoring [3, 42]. We benchmark four state-of-the-art models

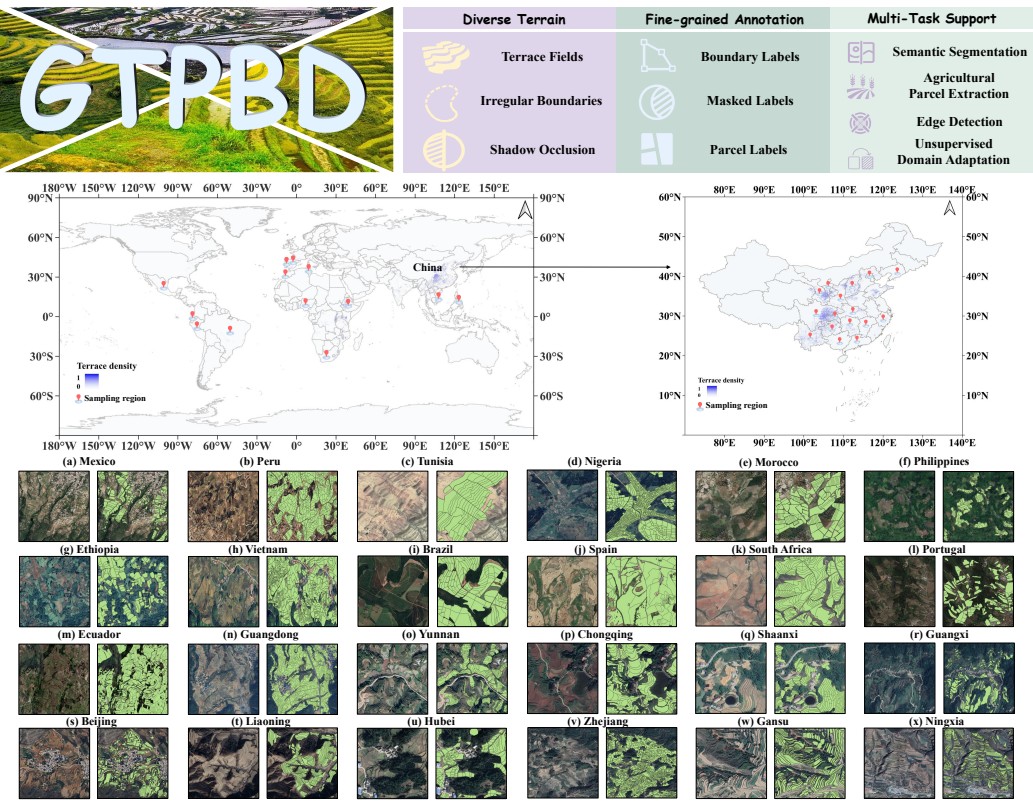

Figure 2: The characteristics of our proposed GTPBD with diverse terrain, fine-grained annotation and multi-task support. GTPBD covers seven major geographic zones in China and transcontinental climatic regions around the world.

on GTPBD: REAUNet-Sober [22], which embeds Sobel-style filters into a U-Net backbone; MuGE [55], employing multi-scale gating units to enhance edge features; PiDiNet [31], a lightweight pixel-difference network optimized for real-time inference; and UEAD [56], leveraging unsupervised edge-aware representation learning. Although these models have demonstrated strong performance on urban or synthetic benchmarks, their efficacy on complex terraced agricultural landscapes has not been evaluated. Our results on GTPBD's fine-grained boundary annotations therefore provide accurate assessment of edge detection performance in terraced farmland environments.

**Unsupervised Domain Adaptation (UDA)** aims to transfer knowledge from a labeled source domain to an unlabeled target domain [33]. UDA methods for remote sensing can be broadly grouped into adversarial training and self-training approaches. Adversarial training such as Cycada [15] and CLAN [23] leverage discriminators to align feature distributions, they incur complex optimization overhead and may not directly address style discrepancies. Representative self-training frameworks include DAFormer [16], which integrates rare-class sampling and pseudo-label denoising. In addition, input-level adaptation methods like FDA [43] perform only low-frequency transfer in the Fourier domain, requiring no adversarial network or pseudo label. Although these UDA methods have been widely used in remote sensing community with some improvements, existing public remote sensing datasets have been limited by urban scenes [20, 7] or land cover mapping [38, 53]. To this end, the GTPBD dataset is proposed for a more challenging benchmark, promoting future research of cross-regional or temporal agricultural and terraced parcel extraction algorithms and its applications.

## 3 Dataset

### 3.1 Image collection and manual annotation

We collect our images in our dataset based on the Global Terrace Map (GTM) with 10-m resolution [21]. To ensure data quality and spatial coverage, we adopted the following screening strategies:

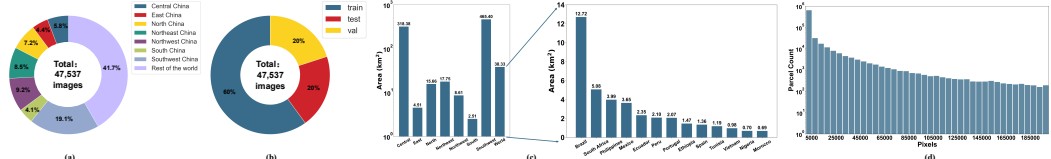

Figure 3: Statistics for the number and area of GTPBD dataset. (a) Distribution of images across different regions. (b) Distribution of images across different dataset splits. (c) Distribution of area across different regions. (d) Distribution of the parcel sizes (logarithmic scale in vertical axis).

**Spatial Coverage and Resolution** Following GTM [21], we prioritized regions with dense terrace distribution and complex topography, and further included representative terraces across diverse landforms—such as alpine terraces in southwestern China and the Sa Pa terraces in Vietnam—to assess model generalization across varied terrains. The dataset spans fourteen countries worldwide, including China, Vietnam, Tunisia, Ethiopia, Peru, and Mexico, as well as seven major geographic regions of China (Central, South, North, East, Northwest, Northeast, and Southwest). High-resolution images were collected from GF-2 and Google Earth with spatial resolutions ranging from 0.1 m to 1 m, covering different seasons and lighting conditions. All images were selected from cloud-free scenes acquired between 2021 and 2025 to ensure consistency and research suitability. Examples are shown in Fig.2, with additional samples in AppendixB.1.

**Parcel Annotation** We construct the topological vectorization and annotations in terraced fields through QGIS. We recruit over 50 participants, comprising undergraduate and graduate students, to manually annotate the fine-grained terraced parcel and boundary with strict quality inspection. The specific implementation strategy is as follows: for areas with dense hierarchical structures and narrow boundaries (less than 0.5m), the common edge annotation method is adopted - prioritizing vectorization of large continuous parcels, and generating sub-parcel topologies through internal line feature cutting; For areas where the field ridge width is greater than or equal to 0.5 m, bilateral segmentation annotation is applied — independent vector boundaries are generated along both sides of the field ridge, and the ridge itself is removed to form a non-standard edge topology structure.

**Three-Level Labels** As for mask labels, we use GDAL's rasterize function for fully connected rasterization (all touched strategy), establish a binary semantic segmentation benchmark, and assign 1 to terraced areas and 0 to backgrounds. As for boundary labels, we use a $3 \times 3$ rectangular grid mask to perform a single morphological etching and construct a 3-pixel-wide edge detection dedicated label. As for parcel labels, we generate pure parcel regions through mask boundary using the XOR operation to achieve terraced parcel extraction tasks. Appendix B.2 displays the differences between previous dataset and our GTPBD.

**Dataset Construction** To address potential spatial correlation or leakage between patches, we first resampled all source images and corresponding labels to a unified resolution of 0.5–0.7 m. Images from the same geographic region share the same target resolution and were split into training (60%), validation (20%), and testing (20%) sets *before* cropping, ensuring spatial independence between subsets. Following common practice in high-resolution agricultural benchmarks such as AI4Boundaries[10], Agriculture-Vision [9], and FUSU [45], we adopted a patch size of $512 \times 512$ pixels. Compared to $256 \times 256$ patches, this size preserves more complete geometric patterns and semantic context, such as parcel shapes and boundaries.

### 3.2 Statistics

This section will introduce some statistics of the GTPBD dataset. With the collection of Global Terrace Map (GTM) [21], the number of labeled images from different regions has been count and the images belonging to each region are divided into training, testing and validation sets with the same scale structure. As is shown in the Fig. 3 (a), our proposed GTPBD dataset contains a large number of images from Southwest China which is followed by Rest of the world, which reflects the global distribution of terraces well. About dataset splits, the majority of the images (60%) are allocated to the training set, while the testing and validation sets each contain 20% of the total images (Fig. 3 (b)). Fig. 3 (c) shows a detailed construction of the spatial coverage across various regions, which suggests that GTPBD dataset is collected mainly from Southwest China, Central China and other parts of the world. Fig. 3 (d) illustrates the parcel size distribution, and because logarithmically scaled vertical coordinate is used, it can be seen that the majority of terraced parcels are smaller than

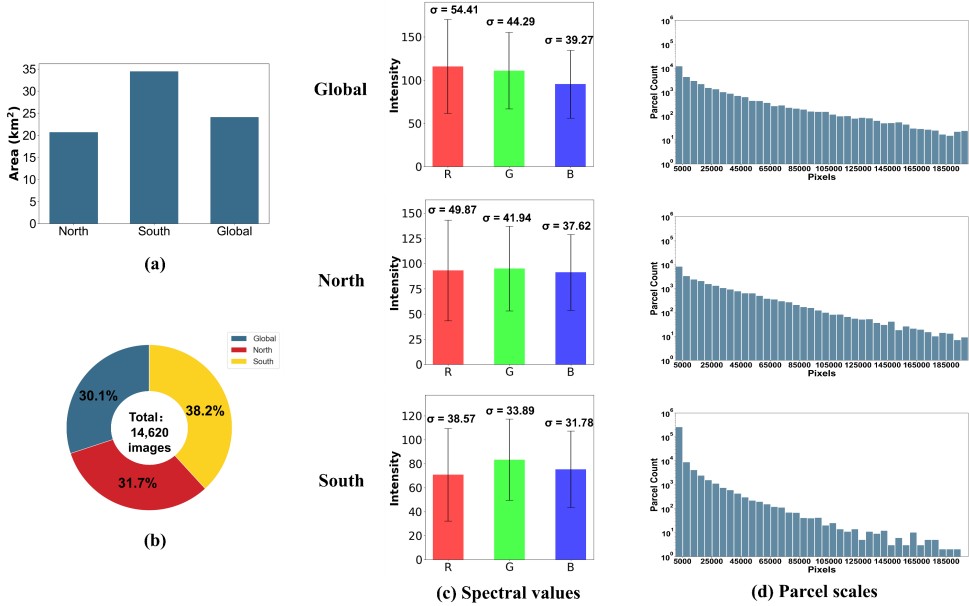

Figure 4: Statistics for three different domains. (a) Distribution of area across different domains. (b) The number of images across different domains. (c) Spectral statistics of mean and standard deviation ($\sigma$) for different domains. (d) Distribution of parcel sizes across different domains (logarithmic scale).

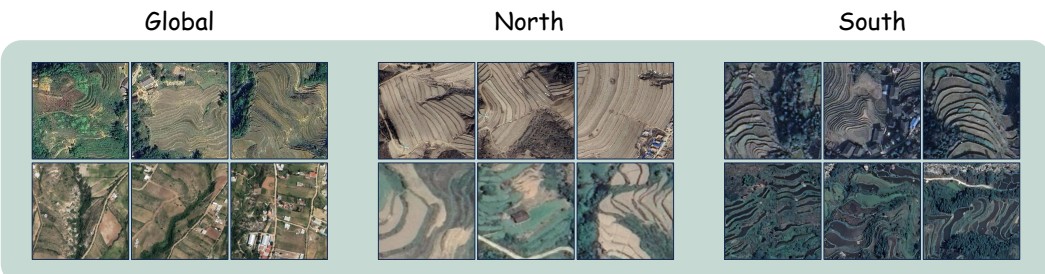

Figure 5: Some cases for three different domains: **Global**, **North** and **South**.

5000 pixels, which presents a challenge to the parcel extraction task. More regional statics of terraced parcel size could be found in Appendix B.3.

### 3.3 Differences among three domains

We seperate our GTPBD into three domains: South China (**South**), North China (**North**) and other regions except China (**Global**). The detailed statics of these three domains are shown in Fig. 4 (a) and (b). According to Fig. 5, the characteristics from different regions are really different.

For the spectral statistics, the mean values are similar (Fig. 4 (c)) because of the large-scale homogeneous geographical areas and diverse land cover types. We could observe that the **South** domain have lower standard deviations. As is shown in Fig. 4 (d), although all domains exist "long tail" phenomenon (logarithmic scale in vertical axis), the **South** domain has the most of the parcels have relatively small scales. When faced with large-scale terraced parcel and boundary extraction tasks, the differences between different scenes or regions bring new challenges to the model generalization and transferability.

Table 2: Comparisons of different semantic segmentation methods on the GTPBD dataset (%). More visualization results and the performance of regions could be found in Appendix E.1.3 and E.1.2.

| Method | Prec.↑ | Rec.↑ | IoU↑ | OA↑ | F1-score↑ |
|---|---|---|---|---|---|
| UNet [29] | 74.11 | 54.93 | 46.09 | 75.46 | 63.09 |
| DeepLabV3 [6] | 69.64 | 73.45 | 57.04 | 78.28 | 71.58 |
| PSPNet [52] | 68.33 | 72.41 | 54.22 | 76.65 | 72.31 |
| Nonlocal [40] | **75.06** | 70.27 | 51.48 | **79.52** | 72.58 |
| OCRNet [46] | 73.05 | 71.56 | 53.87 | 76.95 | 72.30 |
| K-Net [50] | 74.56 | 61.04 | 50.52 | 77.17 | 67.13 |
| SegFormer [41] | 74.45 | 69.07 | 55.84 | 78.14 | 71.66 |
| Mask2Former [8] | 71.22 | **74.33** | **57.16** | 78.73 | **72.74** |

Table 3: Comprehensive performance comparison of edge detection models on the GTPBD dataset.

| Method | ODS↑ | OIS↑ | AP↑ |
|---|---|---|---|
| MuGE [55] | 62.56 | 61.93 | 65.12 |
| PiDiNet [31] | 53.70 | 53.12 | 52.92 |
| UEAD [56] | 25.88 | 26.01 | 17.94 |
| REAUNet-Sober [22] | **65.06** | **63.73** | **70.09** |

# 4 Experiments

## 4.1 Experimental setup

The data splits followed Sec. 3.1. During the training, we used the Stochastic Gradient Descent (SGD) optimizer with a momentum of 0.9 and a weight decay of $10^{-4}$. For the data augmentation, $512 \times 512$ patches were randomly cropped with random mirroring and rotation. We implemented all our experiments on NVIDIA RTX 4090 GPU. We benchmark the GTPBD dataset on eight semantic segmentation methods (U-Net [29], DeepLabV3 [6], PSPNet [52], NonLocal [40], OCRNet [46], K-Net [50], SegFormer [41] and Mask2Former [8]), four edge extraction methods (MuGE [55], PiDiNet [31], UEAD [56] and REAUNet-Sober [22]), three parcel extraction methods (REAUNet [22], SEANet [19], HBGNet [51]) and five UDA methods (Source only, FDA [43], PiPa [24], HRDA [17] and DAFormer [16]). More implementation details are provided in the Appendix C.

## 4.2 Evaluation metrics

We brief list our comprehensive evaluation metrics for different tasks including pixel-level, object-level and edge detection of GTPBD. More details could be found in Appendix D.

(1) **Pixel-level evaluation metrics:** To evaluate segmentation accuracy on the GTPBD dataset, we adopt five standard pixel-level metrics commonly used in semantic segmentation and unsupervised domain adaptation (UDA) tasks, including Precision (**Prec.**), Recall (**Rec.**), Intersection over Union (**IoU**), Overall Accuracy (**OA**) and **F1-score**; (2) **Object-level geometric metrics:** To evaluate the geometric accuracy of parcel delineation on the GTPBD dataset, we adopt three object-level metrics: Global Over-Classification Error (**GOC**), Global Under-Classification Error (**GUC**), and Global Total Classification Error (**GTC**). These indicators provide a comprehensive assessment of shape fidelity and segmentation precision at the object level; (3) **Edge detection evaluation metrics:** For edge detection tasks on the GTPBD dataset, we evaluate model performance using three standard metrics: Optimal Dataset Scale F1-score (**ODS**), Optimal Image Scale F1-score (**OIS**), and Average Precision (**AP**). These metrics jointly assess the adaptability and stability of boundary predictions.

## 4.3 Semantic Segmentation Results

Semantic segmentation plays a vital role in agricultural parcel extraction. As summarized in Table 2, these models exhibit notable discrepancies in segmentation performance, reflecting varying capabilities in modeling fine-grained agricultural boundaries. Specifically, the NonLocal model achieved the highest Prec. (75.06%) and OA (79.52%), indicating strong discrimination with minimal false positives. In contrast, Mask2Former demonstrated superior generalization, attaining the best scores in Rec. (74.33%), IoU (57.16%), and F1-score (72.74%). These findings highlight a trade-off between precision and recall among different architectures and emphasize the robustness of transformer-based models under complex terraced scenarios. Fig. 6 visually compare the segmentation results on a sample patch, with red highlighting false positives and blue highlighting false negatives. As can be observed, conventional semantic segmentation can only distinguish farmland regions from non-farmland areas, but fails to delineate precise parcel boundaries — especially for adjacent parcels that share common borders. More visualization results and the performance of regions could be found in Appendix E.1.

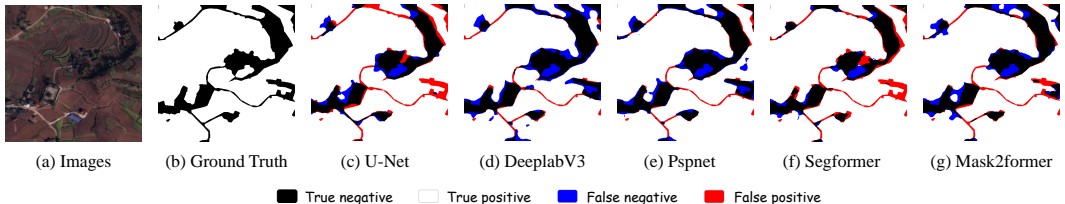

(a) Images    (b) Ground Truth    (c) U-Net    (d) DeeplabV3    (e) Pspnet    (f) Segformer    (g) Mask2former

■ True negative    □ True positive    ■ False negative    ■ False positive

Figure 6: Qualitative comparisons of semantic segmentation and error maps among different methods.

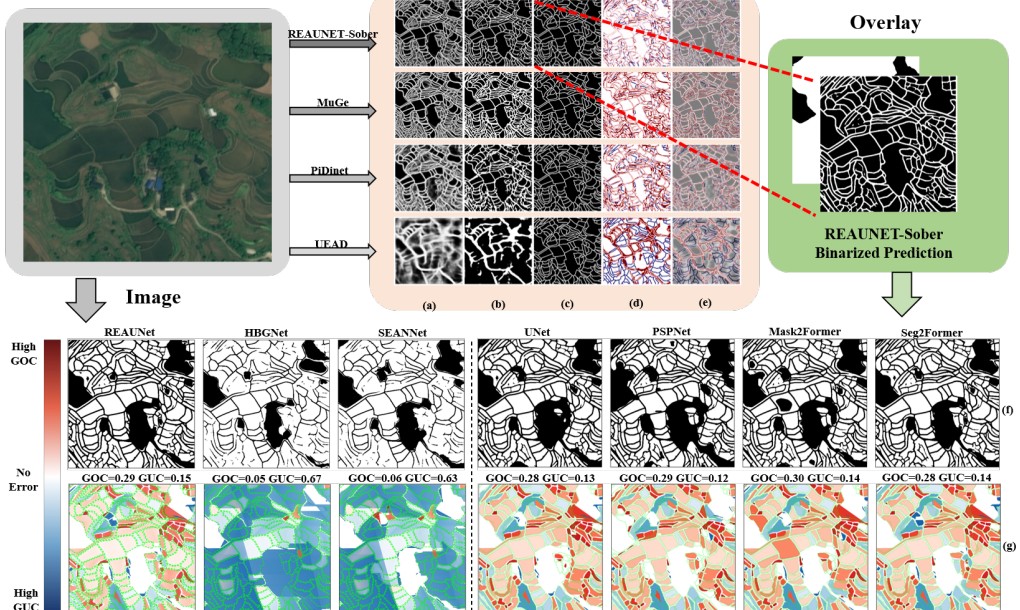

Figure 7: Edge detection and terraced parcel extraction. (a) Probability Map. (b) Binarized Prediction. (c) Ground Truth. (d) Error Regions (Red: FP, Blue: FN). (e) Overlay Visualization (alpha=0.5). (f) The performance of integrating edge results (REAUNET-Sober) and semantic segmentation results. (g) Evaluation of object-level parcel extraction (Red: GOC, Blue: GUC, Green: GT boundary).

## 4.4 Edge Detection Results

Conventional edge detection datasets typically include multi-scale, manually annotated boundaries. To simulate this for the GTPBD dataset, we apply morphological operations (erosion and dilation) to generate edge labels with widths of 1 to 5 pixels, mimicking human-labeled multi-resolution boundaries. Table 3 presents the results of four edge detection models. Among them, REAUNet-Sober achieves the best performance across all three metrics—ODS (65.06%), OIS (63.73%), and AP (70.09%)—significantly outperforming both general-purpose (e.g., PiDiNet), demonstrating its effectiveness in delineating parcel edges under noisy conditions. Visual examples in Fig. 7 (d) further illustrate model robustness, where false positives are marked in red and missed detections in blue.More visualization results and the performance of regions could be found in AppendixE.2

## 4.5 Boundary-integration terraced parcel extraction results

Building on the boundary constraint theory proposed by Yuan et al. [47], which highlights the importance of edge orientation and intensity in enhancing object-level delineation, we explore boundary-integration strategies for agricultural parcel segmentation. We compare four general-purpose segmentation models (U-Net, PSPNet, SegFormer, and Mask2Former) with three agriculture-specific ones (HBGNet, REAUNet, and SEANet). To incorporate edge cues, we introduce a boundary detection pipeline (see Fig. 7) that explicitly integrates edge information into the segmentation process. REAUNet-Sober achieves the best performance among the trained edge detectors, and its binarized output is used to guide segmentation refinement. As shown in Table 4, Mask2Former delivers the best object-level performance, achieving the lowest GOC (22.04%) and GTC (35.53%), despite lower

Table 4: Performance Comparison of Parcels Extraction(unit:%)

| Method | Prec.↑ | Rec.↑ | IoU↑ | OA↑ | F1↑ | GOC↓ | GUC↓ | GTC↓ |
|---|---|---|---|---|---|---|---|---|
| U-Net [29] | **74.34** | 54.29 | 45.72 | 75.39 | 62.75 | 43.57 | **37.04** | 40.43 |
| PSPNet [52] | 68.55 | 71.50 | 53.84 | 76.59 | 69.99 | 22.66 | 49.90 | 39.75 |
| SegFormer [41] | 74.71 | 68.23 | 55.43 | 79.05 | 71.32 | 27.15 | 41.17 | 35.84 |
| Mask2Former [8] | 71.48 | 73.42 | 56.79 | 78.66 | 72.44 | **22.04** | 45.15 | **35.53** |
| SEANet [19] | 70.69 | 79.56 | 59.89 | 76.07 | 74.04 | 26.80 | 54.70 | 44.08 |
| REAUNet [22] | 73.03 | 78.01 | 60.56 | 80.60 | 75.44 | 27.02 | 42.25 | 36.07 |
| HBGNet [51] | 72.50 | **81.81** | **62.44** | **81.20** | **76.88** | 27.40 | 42.52 | 35.79 |

Table 5: Comparisons of UDA and Boundary-integration UDA Performance on GTPBD (%).

| Domain | Method | Conventional UDA | | | | | Boundary-integration UDA (Integrated by REAUNet-Sober) | | | | | | | |
|---|---|---|---|---|---|---|---|---|---|---|---|---|---|---|
| | | Prec.↑ | Rec.↑ | IoU↑ | OA↑ | F1↑ | Prec.↑ | Rec.↑ | IoU↑ | OA↑ | F1↑ | GOC↓ | GUC↓ | GTC↓ |
| S → N | Source Only | 61.57 | 68.75 | 48.11 | 71.07 | 64.96 | 61.55 | 68.58 | 48.01 | 71.03 | 64.88 | 25.49 | 49.12 | 39.13 |
| | FDA[43] | 75.60 | 46.72 | 40.60 | 73.33 | 57.75 | 75.59 | 46.41 | 40.36 | 73.25 | 57.51 | 53.95 | 34.20 | 45.16 |
| | PiPa [24] | 62.74 | **84.71** | **56.35** | 74.41 | **72.09** | 62.72 | **84.05** | **56.05** | 74.29 | **71.84** | 13.79 | 54.71 | 40.91 |
| | HRDA [17] | 81.48 | 59.31 | 52.26 | **78.87** | 68.65 | 81.56 | 58.78 | 51.88 | **78.73** | 68.32 | 42.02 | 26.27 | **35.25** |
| | DAFormer [16] | **82.12** | 58.18 | 51.64 | 78.74 | 68.11 | **82.15** | 57.79 | 51.35 | 78.64 | 67.85 | 41.08 | **29.49** | 35.78 |
| S → G | Source Only | 66.26 | 72.53 | 52.97 | **77.63** | 69.25 | 66.27 | 72.46 | 52.94 | 77.63 | 69.23 | 26.62 | 48.39 | 39.95 |
| | FDA[43] | **67.03** | 53.79 | 42.54 | 74.76 | 59.68 | **67.21** | 52.71 | 41.93 | 74.64 | 59.08 | 48.20 | 44.08 | 46.80 |
| | PiPa [24] | 61.33 | **80.99** | **53.61** | 75.66 | **69.80** | 61.54 | **79.33** | **53.03** | 75.60 | **69.31** | 18.06 | 54.68 | 40.71 |
| | HRDA [17] | 62.85 | 73.72 | 51.35 | 75.74 | 67.85 | 63.15 | 72.01 | 50.70 | 75.69 | 67.29 | 22.48 | 51.45 | **39.70** |
| | DAFormer [16] | 66.24 | 62.43 | 47.36 | 75.90 | 64.28 | 66.43 | 61.36 | 46.84 | 75.81 | 63.79 | 38.18 | **42.73** | 40.51 |
| N → S | Source Only | **82.18** | 68.18 | 59.40 | 81.21 | 74.53 | **82.36** | 68.03 | 59.38 | 81.23 | 74.51 | 32.40 | **33.53** | 33.46 |
| | FDA[43] | 71.37 | 62.66 | 50.07 | 74.81 | 66.73 | 71.62 | 62.05 | 49.80 | 74.78 | 66.49 | 40.43 | 50.03 | 45.48 |
| | PiPa [24] | 79.28 | 80.71 | 66.65 | 83.72 | 79.99 | 79.55 | 79.91 | 66.29 | 83.62 | 79.73 | 23.88 | 39.95 | 32.91 |
| | HRDA [17] | 77.19 | **84.01** | **67.30** | **83.74** | **80.46** | 77.51 | **83.13** | **66.98** | **83.77** | **80.22** | 19.23 | 41.83 | **32.55** |
| | DAFormer [16] | 78.26 | 82.41 | 67.05 | 83.67 | 80.28 | 78.59 | 81.56 | 66.73 | 83.60 | 80.05 | 19.39 | 42.21 | 32.85 |
| N → G | Source Only | 73.83 | 56.90 | 47.36 | 78.03 | 64.27 | 73.86 | 56.86 | 47.33 | 78.03 | 64.25 | 45.92 | 33.78 | 40.30 |
| | FDA[43] | 66.10 | 51.38 | 40.66 | 73.96 | 57.82 | 66.22 | 50.68 | 40.27 | 73.89 | 57.42 | 56.64 | 48.15 | 52.57 |
| | PiPa [24] | 69.60 | 80.67 | 59.65 | 81.05 | 74.72 | 69.85 | 79.30 | 59.08 | 80.93 | 74.28 | 24.34 | 42.21 | 34.45 |
| | HRDA [17] | 70.66 | **83.45** | **61.98** | **82.22** | **76.52** | 70.97 | **81.90** | **61.35** | **82.08** | **76.04** | 20.67 | 43.05 | **32.37** |
| | DAFormer [16] | **74.96** | 67.77 | 55.26 | 80.95 | 71.19 | **75.21** | 66.54 | 54.57 | 80.76 | 70.61 | 32.33 | **33.14** | 32.74 |
| G → S | Source Only | **80.06** | 72.26 | 61.24 | 81.56 | 75.96 | **80.22** | 72.10 | 61.22 | 81.58 | 75.94 | 28.63 | **37.75** | 33.50 |
| | FDA[43] | 66.15 | 51.92 | 41.02 | 69.90 | 58.18 | 66.37 | 51.42 | 40.79 | 69.91 | 57.95 | 46.02 | 49.66 | 47.88 |
| | PiPa [24] | 78.06 | **77.84** | **63.86** | 82.23 | **77.95** | 77.96 | **76.65** | 62.99 | 81.84 | **77.30** | 22.67 | 41.10 | **33.29** |
| | HRDA [17] | 74.59 | 76.28 | 60.54 | 79.96 | 75.42 | 74.90 | 75.45 | 60.23 | 79.91 | 75.18 | 25.91 | 42.75 | 35.35 |
| | DAFormer [16] | 78.72 | 76.75 | 61.56 | **82.26** | 77.72 | 79.01 | 76.01 | **63.23** | 82.18 | 77.48 | 26.46 | 40.55 | 34.04 |
| G → N | Source Only | 77.03 | 69.08 | 57.28 | **79.90** | 72.84 | 77.03 | 69.03 | 57.25 | **79.89** | 72.81 | 32.14 | **35.97** | **34.10** |
| | FDA[43] | 53.77 | 75.31 | 45.72 | 65.12 | 62.75 | 53.72 | 74.87 | 45.51 | 65.04 | 62.56 | 25.75 | 56.99 | 44.22 |
| | PiPa [24] | 62.13 | **88.77** | **57.60** | 74.51 | **73.10** | 62.12 | **88.10** | **57.31** | 74.40 | **72.86** | 10.14 | 54.53 | 39.22 |
| | HRDA [17] | 58.89 | 83.01 | 52.56 | 70.77 | 68.90 | 58.85 | 82.40 | 52.28 | 70.66 | 68.66 | 20.83 | 51.16 | 39.01 |
| | DAFormer [16] | 70.80 | 72.87 | 56.03 | 77.69 | 71.82 | 70.81 | 72.26 | 55.68 | 77.56 | 71.53 | 28.16 | 41.19 | 35.28 |
| Avg | Source Only | 73.49 | 68.12 | 54.39 | 77.23 | 70.51 | 73.53 | 68.01 | 54.29 | 77.24 | 70.44 | 31.70 | 42.59 | 36.74 |
| | FDA[43] | 70.00 | 56.13 | 43.25 | 71.99 | 62.22 | 70.22 | 55.94 | 42.92 | 71.92 | 61.91 | 45.17 | 47.19 | 47.10 |
| | PiPa [24] | 69.02 | **82.12** | **59.62** | 78.60 | **74.38** | 69.29 | **81.22** | **59.13** | 78.46 | **74.07** | 17.15 | 47.88 | 36.91 |
| | HRDA [17] | 70.78 | 76.63 | 58.50 | 78.45 | 73.30 | 71.16 | 75.61 | 57.91 | 78.34 | 72.95 | 25.19 | 42.75 | **35.93** |
| | DAFormer [16] | **75.48** | 69.90 | 56.82 | **79.72** | 72.40 | **75.63** | 69.25 | 56.23 | **79.58** | 72.22 | 31.10 | **38.22** | 36.20 |

pixel-level scores. In contrast, HBGNet attains the highest pixel-level accuracy (Recall: 81.81%, IoU: 62.4%, OA: 82.20%, F1: 76.88%) but suffers from higher object-level errors, indicating a trade-off between pixel precision and structural integrity.

## 4.6 Unsupervised domain adaptation results

We evaluate domain robustness of segmentation models through UDA experiments across three distinct domains in GTPBD (see Sec. 3.3). The Source only and other four domain adaptation methods are benchmarked: FDA, DAFormer, HRDA, and PiPa. As shown in Table 5, across all six domain adaptation directions, PiPa achieves the best performance in pixel-level metrics (Rec., IoU, F1), demonstrating stable cross-domain generalization. Notably, HRDA performs best when the source domain is N, achieving superior results across nearly all metrics under this setting. After adding boundary results (integrated by the performance of REAUNet-Sober), further achieves the lowest GOC and GTC, indicating a stronger ability to preserve parcel boundaries and maintain object completeness, while DAFormer attains the lowest GUC, reflecting its advantage in reducing

under-segmentation and improving boundary coherence.Visualization comparisons are provided in Appendix E.4.

## 5 Conclusion

In this paper, we propose a more challenging terraced parcel dataset named **GTPBD** (**G**lobal **T**erraced **P**arcel and **B**oundary **D**ataset), which is the first fine-grained dataset covering major worldwide terraced regions with more than 200,000 complex terraced parcels with manually annotation. GTPBD comprises 47,537 high-resolution images with three-level labels, including pixel-level boundary labels, mask labels, and parcel labels. It covers seven major geographic zones in China and transcontinental climatic regions around the world. Our proposed GTPBD dataset is suitable for four different tasks, including semantic segmentation, edge detection, terraced parcel extraction and Unsupervised Domain Adaptation (UDA) tasks. Accordingly, we benchmark the GTPBD dataset on eight semantic segmentation methods, four edge extraction methods, three parcel extraction methods and five UDA methods, along with a multi-dimensional evaluation framework integrating pixel-level and object-level metrics. Results highlight the limitations of current models in extracting fine-grained terraced parcels and boundaries, especially under domain shifts. Therefore, our proposed GTPBD fills a critical gap in remote sensing benchmarks and provides a critical infrastructure for complex agricultural terrain analysis and cross-scenario knowledge transfer.

## Acknowledgments

This work was supported in part by the National Natural Science Foundation of China under Grant T2125006 and Grant 42401415; in part by the Shenzhen Science and Technology Program under Grant KCXFZ20240903093759004 and Grant KJZD20230923115106012; in part by the Fundamental Research Funds for the Central Universities, Sun Yat-sen University, under Project 24xkjc002; and in part by the Jiangsu Innovation Capacity Building Program under Project BM2022028.

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

# GTPBD: A Fine-Grained Global Terraced Parcel and Boundary Dataset (Supplementary material)

## Table of Contents in Appendix

# A  Broader impact

The proposed GTPBD dataset provides the first large-scale, fine-grained benchmark for terraced parcel analysis across diverse global terrains. Its release is intended to facilitate progress in computer vision methods for agricultural mapping, with potential societal benefits including improved food security analysis, precision agriculture, and climate-adaptive land management. By focusing on underrepresented mountainous regions, GTPBD promotes algorithmic development that extends beyond conventional flat farmland, contributing to more inclusive and geographically equitable AI research.This dataset may assist governmental and environmental agencies in monitoring land use, identifying erosion risks, and planning sustainable agricultural practices. It may also support digital infrastructure for land ownership registration in developing regions, where documentation remains limited or inaccessible.However, the deployment of models trained on GTPBD in real-world scenarios must be approached with care. Potential negative impacts include misuse of parcel delineation algorithms for land commodification, surveillance, or disempowerment of local communities, especially in regions with contested or informal land tenure. Furthermore, any bias introduced during data annotation or model training could disproportionately affect underrepresented regions or farming systems.

To mitigate these risks, we strongly encourage practitioners to collaborate with local stakeholders and domain experts, ensure transparency in model usage, and align applications with ethical land governance principles. The dataset should serve as a scientific tool for advancing equitable and sustainable land analysis, not for enabling exploitative practices.

# B  More image cases

## B.1  More image cases in GTPBD

GTPBD covers seven major geographic zones in China and transcontinental climatic regions around the world. There are different terraces in different regions, and Fig. 8 supplements the images of terraces of the 12 regions mentioned in Fig. 2 . The dataset's comprehensive labeling system features three distinct levels of annotation, with detailed examples illustrated in Fig. 9.

## B.2  Some cases in previous datasets

Fig. 10 displays typical annotation–imagery pairs from six widely used parcel datasets: (a) FHAPD [51], (b) FTW [18], (c) AI4Boundaries [10], (d) PASTIS [13], (e) CP-Set [49], and (f) GFSAD30 [32]. Although each benchmark has advanced parcel delineation research, several common limitations are evident. FHAPD and CP-Set rely on binary masks with coarse edges, leaving thin ridges either over-smoothed or missing. FTW and PASTIS are built on Sentinel-2 imagery (10 m GSD), so images appear blurry and smallholder fields are merged. AI4Boundaries depicts large, orthogonal parcels from mechanised farms, providing little shape diversity. GFSAD30, at 30 m resolution, captures only blocky field outlines and omits fine-scale geometry altogether. These issues—low spatial resolution, overly regular parcel shapes, and imprecise or incomplete boundaries limit the evaluation of models on fragmented, irregular terrains.

## B.3  More statistics of GTPBD

In Sec. 3.2, detailed statistical analyses of the data have been carried out. To clearly illustrate the distribution of parcel sizes across the eight regions, Fig. 11 provides individual bar charts for each area. These charts highlight that small-sized terraced blocks dominate in every region.

# C  Model details

## C.1  Semantic segmentation

MMSegmentation is an open-source semantic segmentation toolbox built on the OpenMMLab ecosystem and PyTorch [2]. It implements a wide range of state-of-the-art architectures, such as U-Net,

---

[2] https://github.com/open-mmlab/mmsegmentation

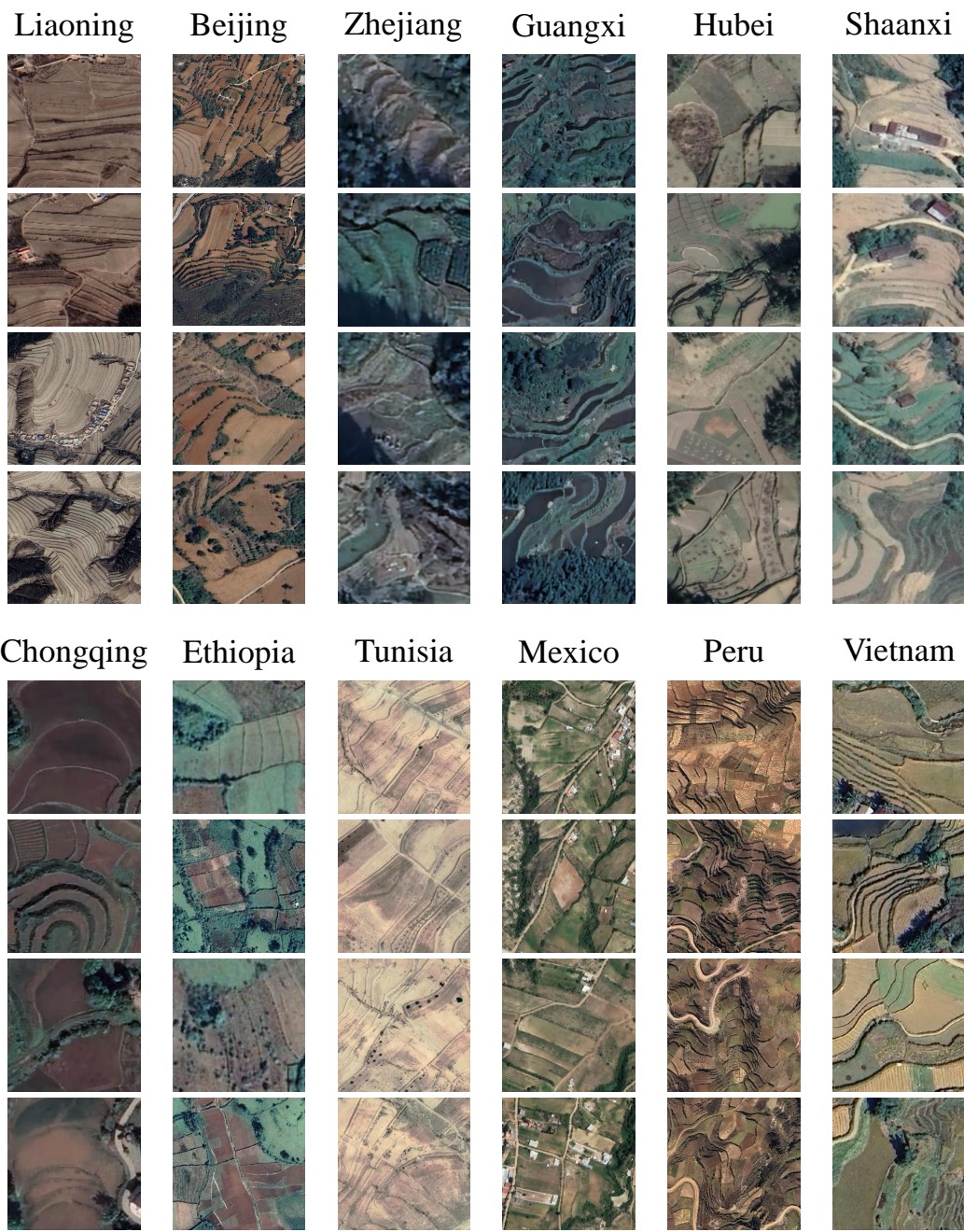

Figure 8: More images cases of different regions in GTPBD.

DeepLabV3, PSPNet, and Segformer, thereby facilitating reproducible comparisons across models. Our semantic segmentation experiments are all completed under the MMsegmention framework.

### C.1.1 U-Net

U-Net, proposed by Ronneberger et al. [29], adopts a symmetric encoder–decoder design that captures contextual information via a contracting path and enables precise localization through an expansive path. In our experiments using MMSegmentation's default U-Net implementation, the key hyperparameters are configured as Table 6.

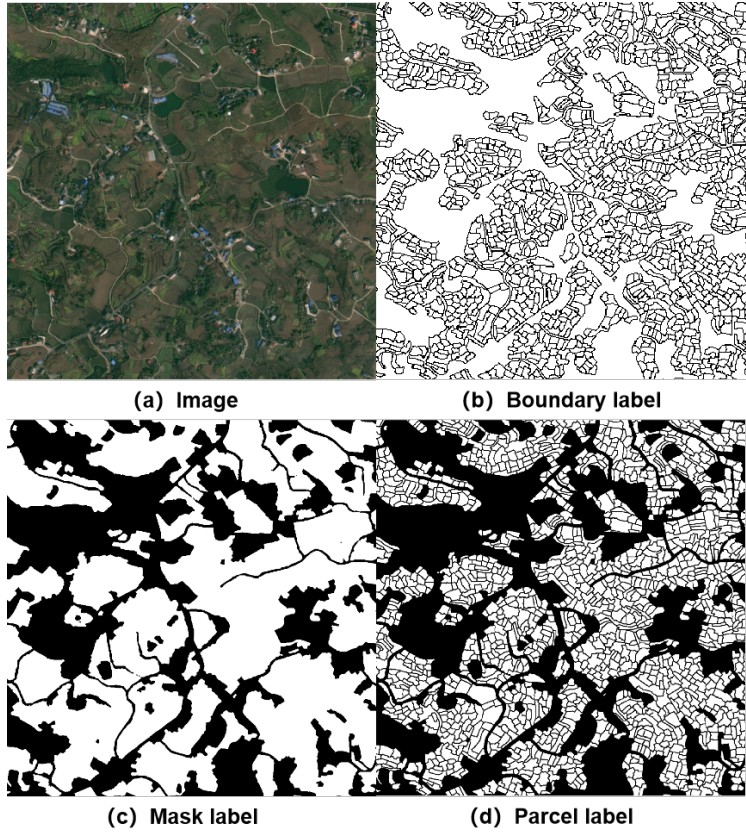

Figure 9: More images cases of three-level labels in GTPBD.

Table 6: The hyperparameters of U-Net

| Hyperparameter | Description | Typical Value |
|---|---|---|
| Input image size | Dimensions of input images | 512×512 |
| Backbone base channels | Number of filters in the first convolution layer | 64 |
| Number of stages | Levels of encoder–decoder (depth of U-Net) | 5 |
| Convolutions per stage | Number of conv layers in each block | 2 |
| Optimizer | Optimization algorithm and initial learning rate | Adam, lr=$1e^{-4}$ |
| Sliding-window crop size | Sliding-window inference patch size | $256 \times 256$ |
| Sliding-window stride | Stride between adjacent sliding-window patches | 170 |
| Training iterations | Total number of training iterations | 20,000 |

### C.1.2   DeepLabV3

DeepLabV3 [6] enhances semantic segmentation by employing atrous (dilated) convolutions to enlarge the receptive field without losing resolution, and by integrating an Atrous Spatial Pyramid Pooling (ASPP) module to capture multi-scale context. In our experiment, All hyper-parameters remain the same as in the mmsegmentation framework and are summarized in Table 7.

### C.1.3   PSPNet

PSPNet [52] introduces a Pyramid Pooling Module that aggregates context information at multiple spatial scales by pooling features into different bin sizes, and then upsampling and concatenating them with the original feature map. This design enables the network to capture both global context

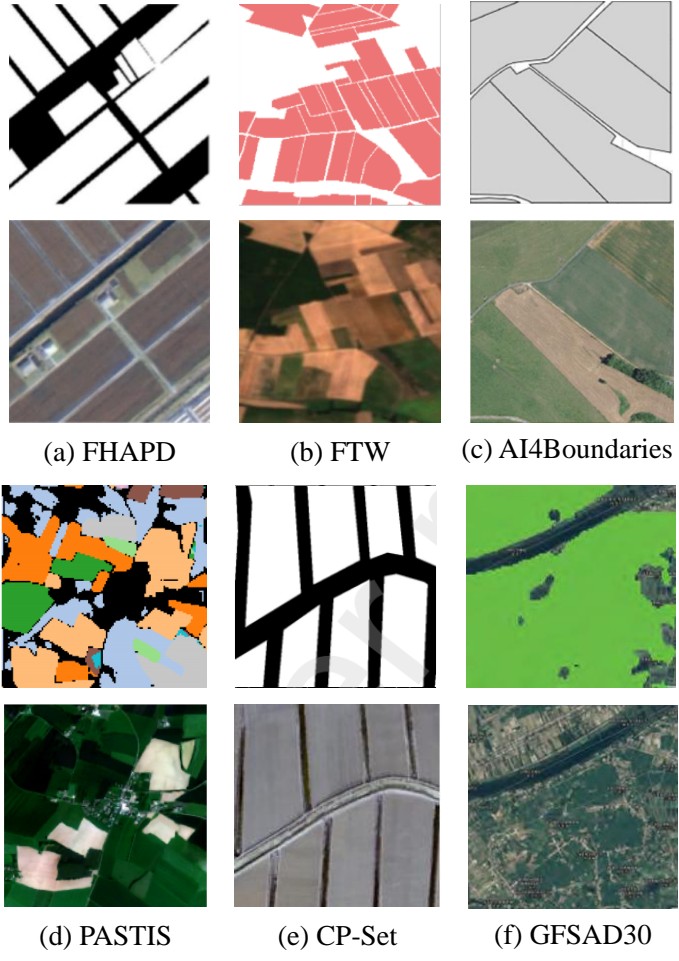



(a) FHAPD       (b) FTW       (c) AI4Boundaries

(d) PASTIS       (e) CP-Set       (f) GFSAD30

Figure 10: Representative Samples from Existing Agricultural Parcel Datasets



and fine details simultaneously. In our experiments, we rely on the out-of-the-box configuration provided by MMSegmentation without further modification, and summarize the key hyperparameters in Table 8.

### C.1.4 NonLocal

Non-local Neural Networks [40], a generic non-local operation that computes pairwise interactions between any two positions on the feature map, enabling the model to capture long-range dependencies and context beyond local receptive fields. We adopt the standard MMSegmentation implementation without modifying its default settings and report the core hyperparameters in Table 9.

### C.1.5 OCRNet

OCRNet [46] enhances pixel-wise segmentation by introducing an object-contextual representation module that aggregates contextual information from object regions to refine per-pixel predictions. Specifically, it first generates a coarse object region map via soft object-region pooling, then computes a pixel-wise contextual representation by weighting features according to their belongingness to these regions. This two-stage design allows the network to capture both local details and global object semantics effectively. Table 10 summarize the main hyperparameters in our experiment.

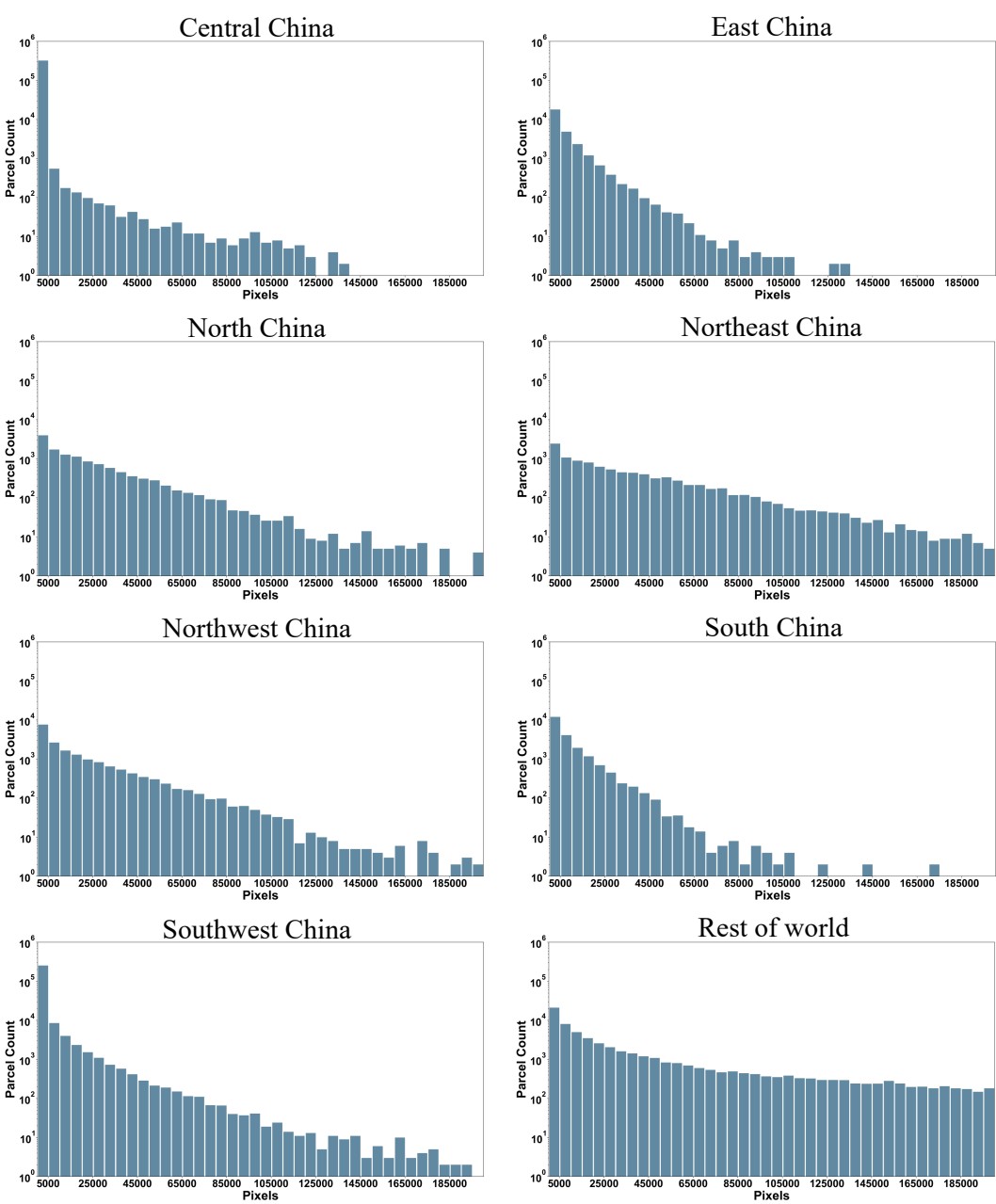

Figure 11: Distribution of parcel sizes across different regions. (logarithmic scale)

### C.1.6 K-Net

K-Net [50] formulates semantic segmentation as a dynamic kernel-based grouping problem, enabling adaptive context aggregation and object-level reasoning. Each kernel generates an attention map that segments the feature map into distinct semantic regions, and these kernels are updated via a lightweight transformer-style interaction. In our experiments, we employ MMSegmentation's K-Net configuration without further modifications and summarize the core hyperparameters in Table 11.

### C.1.7 Segformer

SegFormer [41] leverages a hierarchy of lightweight MLP-based Mix-FFN blocks and multi-level feature fusion to achieve efficient and effective semantic segmentation without positional encodings. Its simple encoder–decoder design employs a series of Transformer-like layers that progressively

Table 7: The hyperparameters of DeeplabV3

| Hyperparameter | Description | Typical Value |
|---|---|---|
| Input image size | Dimensions of input images | 512×512 |
| Backbone architecture | Type and depth of backbone network | ResNetV1c-50 |
| Backbone dilations | Dilation rates for each backbone stage | (1,1,2,4) |
| Backbone strides | Stride sizes for each backbone stage | (1,2,1,1) |
| ASPP dilations | Dilation rates in Atrous Spatial Pyramid Pool | (1,12,24,36) |
| ASPP channels | Number of intermediate channels in ASPP head | 512 |
| Auxiliary head channels | Number of channels in auxiliary FCN head | 256 |
| Number of classes | Number of segmentation target classes | 2 |
| Training iterations | Total number of training iterations | 20,000 |

Table 8: The hyperparameters of PSPNet

| Hyperparameter | Description | Typical Value |
|---|---|---|
| Input image size | Dimensions of input images | 512×512 |
| Backbone architecture | Type and depth of backbone network | ResNetV1c-50 |
| Backbone dilations | Dilation rates for each backbone stage | (1,1,2,4) |
| Backbone stages | Number of residual stages used | 4 |
| PSP pooling scales | Sizes of the pyramid pooling bins | (1,2,3,6) |
| PSP intermediate channels | Number of feature channels after pooling | 512 |
| Auxiliary head channels | Number of channels in auxiliary FCN head | 256 |
| Loss weights | Main and auxiliary segmentation loss weights | 1.0 (main), 0.4 (auxiliary) |
| Training iterations | Total number of training iterations | 20,000 |

downsample the input, and fuse multi-scale representations via a lightweight MLP decoder. Key parameters are shown in Table 12 in our experiment.

### C.1.8 Mask2Former

Mask2Former [8] presents a universal architecture for segmentation tasks by modeling pixel-wise, instance-wise, and semantic-level grouping using a mask-based Transformer decoder. It decouples mask prediction from task-specific heads, employing dynamic queries and multi-scale features to produce segmentation masks across different granularities. We employ Mask2Former configuration directly without any additional tuning and the key hyperparameters can be seen in Table 13.

### C.2 Edge Detection Methods

### C.2.1 UEAD

UEAD [56] addresses the annotation ambiguity and subjectivity inherent in edge detection by modeling label uncertainty. Instead of relying on deterministic pixel-wise labels, it learns a Gaussian distribution over annotations to capture labeling variance. The estimated variance serves as a measure of pixel-level uncertainty, and an adaptive weighting loss emphasizes learning from uncertain (i.e., hard) samples. UEAD can be integrated with various encoder–decoder backbones and consistently improves performance across multiple edge detection benchmarks.

### C.2.2 MuGE

MuGE[55] introduces a novel edge detection framework that captures edge ambiguity by generating edge maps at multiple controllable granularities. It first predicts edge granularity from annotations,

Table 9: The hyperparameters of NonLocal

| Hyperparameter | Description | Typical Value |
|---|---|---|
| Input image size | Dimensions of input images | 512×512 |
| Backbone architecture | Type and depth of backbone network | ResNetV1c-50 |
| Backbone dilations | Dilation rates for each backbone stage | (1,1,2,4) |
| NLhead intermediate channels | Number of feature channels in the NLHead | 512 |
| NL reduction ratio | Channel reduction factor inside the non-local block | 2 |
| NL operation mode | Similarity function used in non-local computation | embedded_gaussian |
| Dropout ratio | Dropout probability applied in the NLHead | 0.1 |
| Training iterations | Total number of training iterations | 20,000 |

Table 10: The hyperparameters of OCRNet

| Hyperparameter | Description | Typical Value |
|---|---|---|
| Input image size | Dimensions of input images | 512×512 |
| Backbone architecture | Type and depth of backbone network | HRNetV2-W18 |
| Number of cascade stages | Number of encoder–decoder stages | 2 |
| FCNhead intermediate channels | Total channels after concatenating multi-scale features | 270 |
| OCRHead channels | Number of feature channels in OCR contextual module | 512 |
| OCR channels | Reduced channel dimension inside OCR module | 256 |
| Training iterations | Total number of training iterations | 20,000 |

then injects this granularity into multi-scale feature maps to produce edge maps ranging from coarse contours to fine textures. By decomposing feature maps into frequency components, MuGE enables fine control over edge detail, resulting in both interpretability and high accuracy.

### C.2.3 PiDiNet

PiDiNet[31] offers a lightweight and efficient solution for edge detection by combining traditional gradient operators with modern convolutional design. It introduces pixel difference convolutions that emulate classical detectors like Canny and Sobel, enabling real-time inference with high accuracy. With less than 1M parameters, PiDiNet achieves human-level performance on BSDS500 and maintains strong efficiency–accuracy trade-offs across diverse benchmarks.

### C.3 Agricultural Parcel Extraction Methods

### C.3.1 REAUNet

REAUNet[22] is a tailored edge-aware network designed for accurate agricultural parcel delineation from both medium- and high-resolution remote sensing imagery (Sentinel-2 and GF-2). It addresses the common issue of unclosed and fragmented parcel edges by integrating four key components: an edge detection block, a dual attention block, a deep supervision mechanism, and a refine module. These modules collectively enhance the model's sensitivity to boundary information and improve multi-scale consistency. Experiments show that all four components are critical, with full REAUNet outperforming partial variants. Compared with SEANet and MPSPNet, REAUNet achieves improvements in both thematic (F1, IoU) and geometric (GTC) accuracy, and demonstrates promising transferability to unseen regions with limited fine-tuning data.

### C.3.2 SEANet

SEANet[19] is a multi-task neural network that simultaneously predicts semantic masks, edges, and distance maps for precise and closed parcel delineation. By explicitly modeling boundary extraction as an edge detection task, SEANet enhances geometric accuracy and is particularly effective for

Table 11: The hyperparameters of K-Net

| Hyperparameter | Description | Typical Value |
|---|---|---|
| Input image size | Dimensions of input images | 512×512 |
| Backbone architecture | Type and depth of backbone network | ResNetV1c-50 |
| Decode head stages | Number of iterative decoding stages | 3 |
| Kernel update FFN channels | Hidden dimension in feed-forward networks of update heads | 2048 |
| Kernel update attention heads | Number of attention heads in each KernelUpdateHead | 8 |
| Convolution kernel size | Kernel size for pointwise conv in update heads | 1 |
| Training iterations | Total number of training iterations | 20,000 |

Table 12: The hyperparameters of Segformer

| Hyperparameter | Description | Typical Value |
|---|---|---|
| Input image size | Dimensions of input images | 512×512 |
| Backbone | Type of hierarchical Transformer encoder | MixVisionTransformer |
| Embed dims | Dimension of the token embeddings at stage 1 | 32 |
| Number of stages | Levels of hierarchical feature extraction | 4 |
| Layers per stage | Number of attention heads in each KernelUpdateHead | 8 |
| Layers per stage | Transformer blocks in each stage | [2,2,2,2] |
| Attention heads | Number of self-attention heads per stage | [1,2,5,8] |
| Patch sizes | Size of convolutional patches at each stage | [7, 3, 3, 3] |
| SR ratios | Spatial reduction ratios before attention | [8,4,2,1] |
| Training iterations | Total number of training iterations | 20,000 |

small, irregular parcels. It incorporates a multi-level edge feature extraction mechanism and a task uncertainty-aware loss to improve generalization. Extensive experiments on both high-resolution (GF-2) and medium-resolution (Sentinel-2) images across China and Europe show that SEANet produces more accurate parcel layouts and demonstrates robust cross-region transferability.

### C.3.3 HBGNet

HBGNet[51] is a hierarchical dual-branch framework designed to fully exploit boundary semantics for robust agricultural parcel extraction. It consists of a core AP extraction branch and an auxiliary boundary branch enhanced by Laplacian-based convolution. To improve adaptability across varying parcel sizes and morphologies, HBGNet integrates global–local context aggregation and boundary-guided fusion modules. It also introduces FHAPD, the first large-scale VHR agricultural parcel dataset in China, to support comprehensive evaluation. HBGNet outperforms eight existing methods across multiple datasets (FHAPD, AI4Boundaries, Sentinel-2) in both attribute and geometric metrics, achieving up to 7.5.

### C.4 Unsupervised Domain Adaptation Methods

### C.4.1 FDA

FDA[43] is a simple yet effective UDA method that reduces domain gaps by aligning the low-frequency components of source and target images in the Fourier domain. The core idea is that low-frequency information in images (e.g., color and style) mainly accounts for the domain shift, while high-frequency components capture structural details relevant to semantic content. By replacing the low-frequency spectrum of source images with that of target images, FDA transfers global appearance cues without altering the semantic layout. This process improves feature-level alignment and enhances segmentation performance in the target domain without requiring architectural changes or additional supervision.

Table 13: The hyperparameters of mask2former

| Hyperparameter | Description | Typical Value |
|---|---|---|
| Input image size | Dimensions of input images | 512×512 |
| Backbone architecture | Feature extractor type and depth | ResNet-50 |
| Number of queries | Number of learnable mask queries | 100 |
| Transformer encoder layers | Number of layers in the Deformable DETR transformer encoder | 6 |
| Transformer decoder layers | Number of layers in the Mask2Former transformer decoder | 9 |
| Classification loss weight | Weight for the classification loss in mask prediction | 2.0 |
| Mask & Dice loss weights | Weights for mask and Dice losses | mask=5.0, dice=5.0 |
| Training iterations | Total number of training iterations | 20,000 |

### C.4.2 DAFormer

DAFormer[16] introduces a Transformer-based architecture for UDA in semantic segmentation, leveraging the superior representation capability of vision Transformers. The model integrates a Transformer encoder with a multi-level, context-aware decoder and employs three essential training strategies: rare class sampling to improve pseudo-labels, ImageNet feature distance regularization to stabilize transfer, and learning rate warmup to reduce overfitting. These components collectively enable DAFormer to effectively adapt to the target domain and learn rare or hard classes. DAFormer achieves significant improvements over previous UDA methods, setting new benchmarks on synthetic-to-real segmentation tasks.

### C.4.3 HRDA

HRDA[17] proposes a multi-resolution UDA framework to balance fine-detail preservation and long-range context modeling in semantic segmentation. It combines small high-resolution image crops—effective for boundary and small object delineation—with large low-resolution crops that retain broader contextual information. A learned scale attention module adaptively fuses these dual-resolution representations. HRDA significantly improves segmentation quality on domain shift benchmarks while maintaining computational efficiency, and is particularly effective for detailed structures under limited GPU memory.

### C.4.4 PiPa

PiPa[24] introduces a unified pixel- and patch-level self-supervised learning framework for UDA in semantic segmentation. Unlike traditional UDA methods that only minimize inter-domain discrepancies, PiPa focuses on enhancing intra-domain consistency by modeling pixel-level intra-class compactness and patch-level context-invariant semantics. This dual-level supervision enables the model to learn more robust and domain-invariant representations. PiPa achieves competitive results on standard UDA benchmarks and can be flexibly combined with other UDA methods to further improve performance without increasing model complexity.

## D   Evaluation metrics

### D.1   Pixel-level evaluation metrics

To evaluate segmentation accuracy on the GTPBD dataset, we adopt five standard pixel-level metrics commonly used in semantic segmentation and unsupervised domain adaptation (UDA) tasks, including Precision (**Prec.**), Recall (**Rec.**), Intersection over Union (**IoU**), Overall Accuracy (**OA**) and **F1-score**.

**Precision (Prec.)**  measures the proportion of correctly predicted agricultural pixels among all pixels predicted as agricultural. A higher precision indicates fewer false positives in land parcel classification.

$$Prec. = \frac{TP}{TP + FP} \tag{1}$$

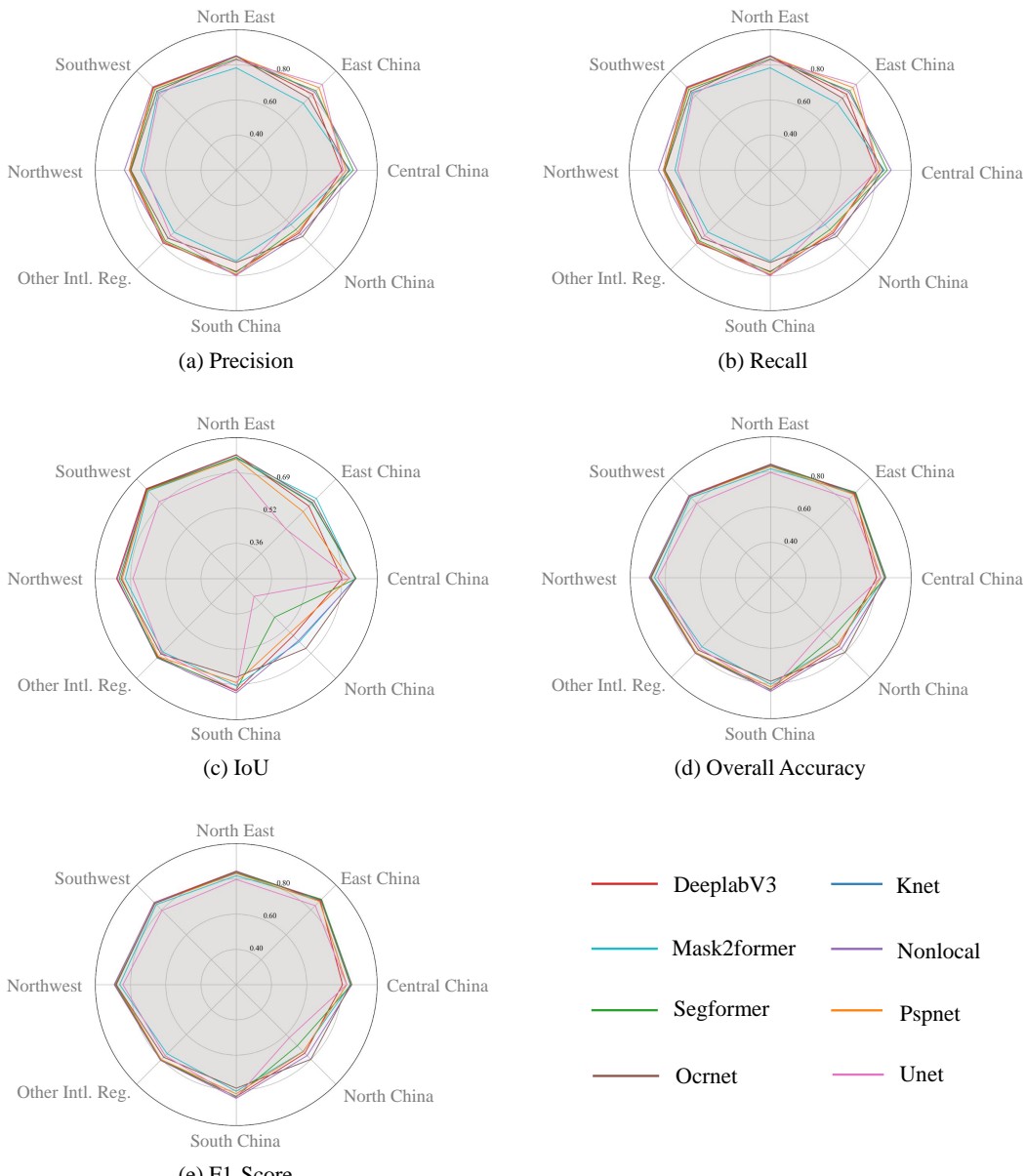

Figure 12: Regional Segmentation Performance Across Models

where TP and FP indicate true positive and false positive, respectively. TP indicates the number of pixels correctly identified as agricultural parcels, while FP indicate the number of pixels mis-identified as agricultural parcels (i.e., mistakes).

**Recall (Rec.)** captures the proportion of true agricultural pixels that are correctly identified. High recall reflects a model's ability to minimize missed detections.

$$Rec. = \frac{TP}{TP + FN} \tag{2}$$

where FN indicates false negative and the number of pixels mis-identified as non-agricultural parcels (i.e., omissions).

**Intersection over Union (IoU)** quantifies the spatial agreement between the predicted and ground-truth regions. IoU is widely adopted in segmentation benchmarks due to its robustness to class

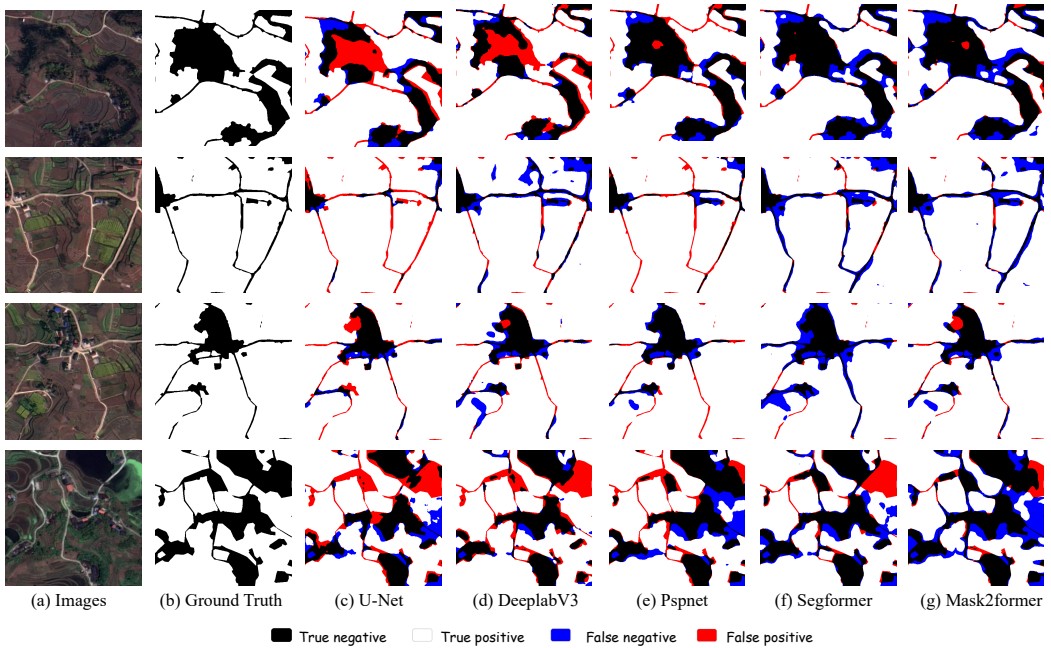

(a) Images    (b) Ground Truth    (c) U-Net    (d) DeeplabV3    (e) Pspnet    (f) Segformer    (g) Mask2former

■ True negative    □ True positive    ■ False negative    ■ False positive

Figure 13: Error Map Comparison of different Semantic Segmentation Models

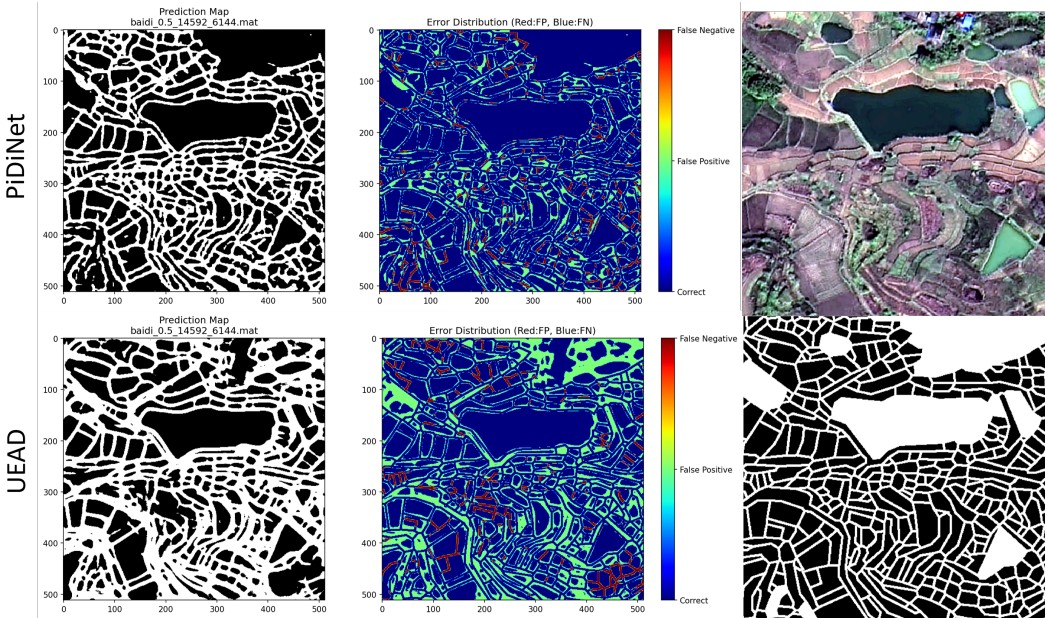

Figure 14: Qualitative edge detection comparisons between PiDiNet [31] and UEAD [56].

imbalance.

$$IoU = \frac{TP}{TP + FP + FN} \tag{3}$$

**Overall Accuracy (OA)** calculates the proportion of correctly classified pixels across the entire image. OA provides a global view of model performance and is especially informative for datasets with class imbalance.

$$OA = \frac{TP + TN}{TP + FP + FN + TN} \tag{4}$$

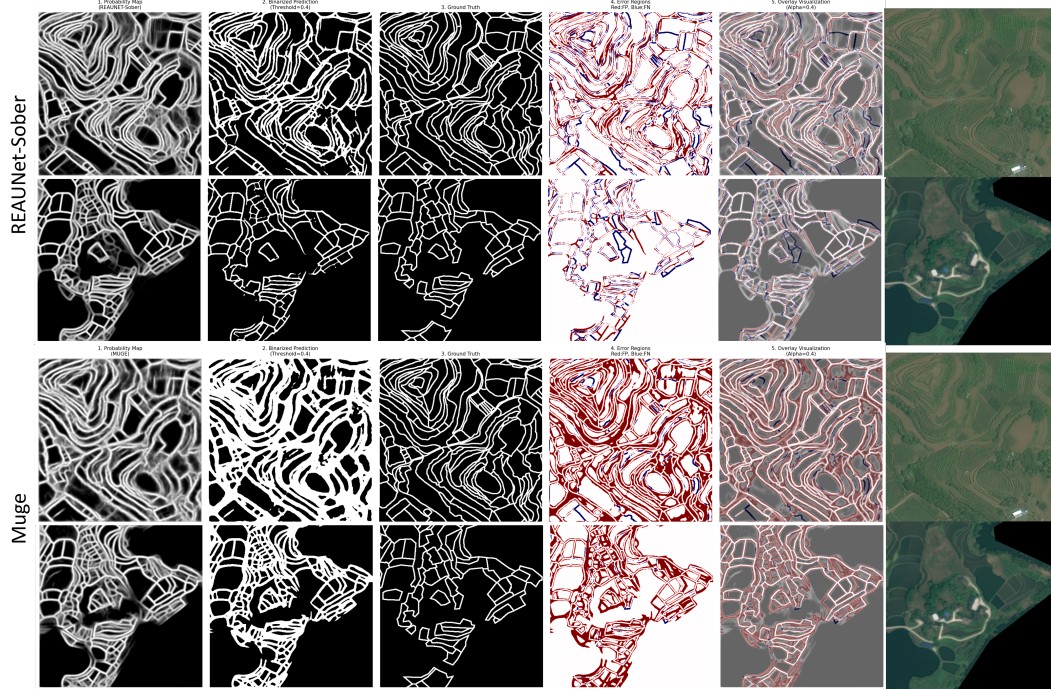

Figure 15: Qualitative edge detection comparisons between REAUNet-Sober [22] and MuGE [55].

where TN indicates true negative and the number of pixels correctly identified as non-agricultural parcels.

**F1-score** is the harmonic mean of precision and recall, offering a balanced evaluation metric particularly useful under domain shift conditions or in presence of noise and uncertainty.

$$F1 = \frac{2 \cdot Prec. \times Rec.}{Prec. + Rec.} \tag{5}$$

### D.2 Object-level geometric metrics

To comprehensively evaluate the geometric quality of delineated agricultural parcels, we adopt three object-level geometric metrics: Global Over-Classification Error (GOC), Global Under-Classification Error (GUC), and Global Total Classification Error (GTC). These indicators quantify segmentation accuracy in terms of spatial overreach, omission, and overall geometric consistency.

Let $S_i$ denote the $i$-th predicted parcel (segmentation), and let $O_i$ represent the ground truth parcel that has the largest intersection area with $S_i$. Denote $m$ as the number of predicted parcels. The object-wise evaluation is defined as follows:

**Global Over-Classification Error (GOC)** measures the average extent to which predicted parcels exceed the spatial extent of their matched ground truth objects:

$$\mathrm{OC}(S_i) = 1 - \frac{\mathrm{area}(S_i \cap O_i)}{\mathrm{area}(O_i)}, \tag{6}$$

$$\mathrm{GOC} = \sum_{i=1}^{m} \left( \mathrm{OC}(S_i) \cdot \frac{\mathrm{area}(S_i)}{\sum_{k=1}^{m} \mathrm{area}(S_k)} \right), \tag{7}$$

where $\mathrm{area}(\cdot)$ denotes the number of pixels in the respective region.

**Global Under-Classification Error (GUC)** quantifies the proportion of each predicted parcel not covered by the corresponding ground truth:

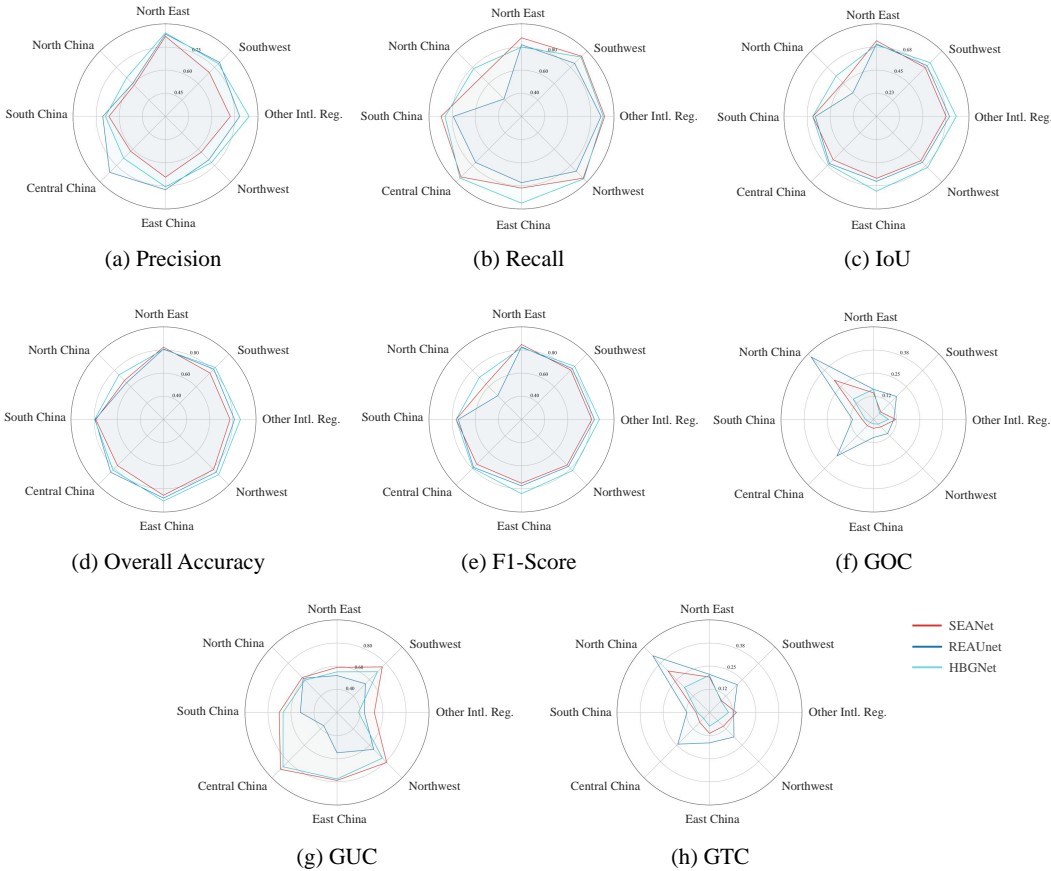

Figure 16: Regional Benchmark of Agricultural Parcel Extraction Models across Eight Metrics

$$\mathrm{UC}(S_i) = 1 - \frac{\mathrm{area}(S_i \cap O_i)}{\mathrm{area}(S_i)}, \tag{8}$$

$$\mathrm{GUC} = \sum_{i=1}^{m} \left( \mathrm{UC}(S_i) \cdot \frac{\mathrm{area}(S_i)}{\sum_{k=1}^{m} \mathrm{area}(S_k)} \right). \tag{9}$$

**Global Total Classification Error (GTC)** synthesizes both over- and under-classification errors into one holistic metric using a root-mean-square formulation:

$$\mathrm{TC}(S_i) = \sqrt{\frac{\mathrm{OC}(S_i)^2 + \mathrm{UC}(S_i)^2}{2}}, \tag{10}$$

$$\mathrm{GTC} = \sum_{i=1}^{m} \left( \mathrm{TC}(S_i) \cdot \frac{\mathrm{area}(S_i)}{\sum_{k=1}^{m} \mathrm{area}(S_k)} \right). \tag{11}$$

### D.3 Edge detection evaluation metrics

For edge detection tasks on the GTPBD dataset, we evaluate model performance using three widely adopted metrics: Optimal Dataset Scale F1-score (ODS), Optimal Image Scale F1-score (OIS), and Average Precision (AP). These metrics jointly assess the accuracy, adaptability, and robustness of predicted boundaries.

Let $P_t$ and $R_t$ denote precision and recall computed at threshold $t$, and let $F_t$ be the corresponding F1-score:

$$F_t = \frac{2 \cdot P_t \cdot R_t}{P_t + R_t}. \tag{12}$$

**Optimal Dataset Scale F1-score (ODS)** evaluates the global performance of an edge detector across the entire dataset using a single optimal threshold $t^*$:

$$\text{ODS} = \max_{t \in \mathcal{T}} \left( \frac{2 \cdot P_t^{\text{dataset}} \cdot R_t^{\text{dataset}}}{P_t^{\text{dataset}} + R_t^{\text{dataset}}} \right), \tag{13}$$

where $P_t^{\text{dataset}}$ and $R_t^{\text{dataset}}$ are aggregated precision and recall over the full dataset under threshold $t$.

**Optimal Image Scale F1-score (OIS)** computes the mean of the per-image best F1-scores, reflecting local threshold adaptiveness:

$$\text{OIS} = \frac{1}{N} \sum_{i=1}^{N} \max_{t \in \mathcal{T}} \left( \frac{2 \cdot P_t^{(i)} \cdot R_t^{(i)}}{P_t^{(i)} + R_t^{(i)}} \right), \tag{14}$$

where $P_t^{(i)}$ and $R_t^{(i)}$ denote the precision and recall on the $i$-th image under threshold $t$, and $N$ is the total number of images.

**Average Precision (AP)** is computed as the area under the precision–recall curve:

$$\text{AP} = \int_0^1 P(R) \, dR, \tag{15}$$

where $P(R)$ is the precision as a function of recall, evaluated across all thresholds.

| Dataset | Model | Pixel-level | | | | | Edge-level | | Object-level | | |
|---|---|---|---|---|---|---|---|---|---|---|---|
| | | Prec.↑ | Rec.↑ | IoU↑ | OA↑ | F1↑ | OIS↑ | ODS↑ | GOC↓ | GUC↓ | GTC↓ |
| | UNet | **91.03** | 95.03 | 86.89 | 90.55 | 92.99 | **81.40** | **70.39** | 5.90 | **41.97** | **29.97** |
| | DeepLabV3 | 90.83 | **97.06** | **88.40** | 91.60 | **93.84** | 66.45 | 56.50 | **2.79** | 51.24 | 36.29 |
| FHAPD[51] | PSPNet | 88.67 | 95.63 | 85.22 | 89.06 | 92.02 | 55.48 | 44.06 | 3.76 | 55.01 | 38.99 |
| | SegFormer | 90.40 | 94.61 | 85.97 | 89.82 | 92.46 | 68.97 | 57.37 | 4.80 | 50.09 | 35.58 |
| | Mask2Former | 90.42 | 94.51 | 85.90 | 89.77 | 92.42 | 74.92 | 64.05 | 5.66 | 44.87 | 31.98 |
| | UNet | 81.50 | 78.19 | 66.40 | 84.64 | 79.81 | 28.75 | 20.86 | 31.34 | 38.92 | 35.33 |
| | DeepLabV3 | 81.08 | 82.31 | 69.05 | 85.67 | 81.69 | 32.51 | 22.05 | 19.46 | 29.43 | 24.95 |
| AI4Boundaries (Ortho)[10] | PSPNet | 83.60 | 82.54 | 71.04 | 86.93 | 83.07 | 36.43 | 24.11 | 21.63 | 34.25 | 28.64 |
| | SegFormer | **87.37** | 78.45 | 70.46 | 87.23 | 82.67 | 29.88 | 20.14 | 20.57 | 26.32 | 23.62 |
| | Mask2Former | 84.58 | **83.49** | **72.46** | **87.68** | **84.03** | **38.44** | **27.04** | **12.76** | **25.80** | **20.35** |
| | UNet | 83.90 | 79.24 | 68.79 | 90.78 | 81.51 | **73.33** | **62.39** | 25.81 | 34.36 | 30.39 |
| | DeepLabV3 | 85.92 | **84.02** | **73.86** | **92.37** | **84.96** | 70.38 | 57.28 | **15.63** | 38.30 | 29.18 |
| FTW[18] | PSPNet | 82.95 | 81.92 | 70.11 | 91.05 | 82.43 | 65.92 | 52.63 | 18.17 | 42.23 | 32.51 |
| | SegFormer | 83.47 | 80.05 | 69.09 | 90.82 | 81.72 | 68.56 | 56.70 | 18.64 | 38.66 | 30.35 |
| | Mask2Former | **86.06** | 77.88 | 69.13 | 91.08 | 81.75 | 72.15 | 61.04 | 20.44 | **30.88** | **26.19** |
| | UNet | **74.11** | 54.93 | 46.09 | 75.46 | 63.09 | 22.47 | 15.17 | 42.69 | **37.27** | 37.84 |
| | DeepLabV3 | 69.64 | 73.45 | 57.04 | 71.58 | 78.28 | 20.08 | 13.53 | 21.38 | 45.59 | 35.61 |
| GTPBD(ours) | PSPNet | 68.33 | 72.41 | 54.22 | 76.65 | 70.31 | 19.66 | 13.08 | 21.58 | 50.06 | 39.21 |
| | SegFormer | 74.45 | 69.07 | 55.84 | 78.14 | 71.66 | 22.97 | 15.70 | 26.10 | 41.44 | **34.63** |
| | Mask2Former | 71.22 | **74.33** | **57.16** | **78.73** | **72.74** | **25.56** | **17.59** | **21.01** | 45.26 | 35.28 |

Table 14: Semantic segmentation results across datasets and models. Pixel-level metrics include Precision, Recall, IoU, Overall Accuracy (OA) and F1-Score. Edge-level metrics include OIS and ODS. Object-level metrics are GOC, GUC, and GTC.

# E   More results

## E.1   More results on Semantic Segmentation

### E.1.1   Results on Multiple Datasets

We provide detailed semantic segmentation results of five representative models (UNet [29], DeepLabV3 [6], PSPNet [52], SegFormer [41], and Mask2Former [8]) on four benchmark datasets: FHAPD [51], AI4Boundaries (Ortho) [10], FTW [18], and the proposed GTPBD dataset. The evaluation considers three categories of metrics, namely pixel-level, edge-level, and object-level, as summarized in Table 14.

Several key observations can be drawn from the results. On GTPBD and AI4Boundaries, Mask2Former achieves the best performance across almost all metrics. On FTW and FHAPD, DeepLabV3 consistently outperforms other models, while UNet also demonstrates competitive boundary detection performance. Notably, GTPBD yields the lowest scores across all models, confirming its higher difficulty and greater diversity compared to existing datasets. These findings highlight that no single model is universally superior across all benchmarks, and that our proposed dataset presents greater challenges for semantic segmentation.

### E.1.2   Discussions the results for different regions

The radar plots in Fig. 12 compare five key metrics—Precision, Recall, IoU, Overall Accuracy, and F1-Score—across seven geographic regions for each segmentation architecture. Overall, all models achieve similar performance profiles, though U-Net demonstrates relatively average performance in terms of IoU and F1-Score, trailing behind advanced transformer-based architectures such as Mask2Former across most regions.

### E.1.3   More visualization cases

Fig. 13 shows error maps for each model on the same test region, where black denotes true negatives, white true positives, blue false negatives, and red false positives. This visualization highlights that transformer-based methods such as Mask2Former [8] produce fewer false negatives in complex boundary areas. However, overall, the models struggle to accurately segment very fine-grained background details.

## E.2   More visualization results on edge detection

Fig. 14 and Fig. 15 present qualitative comparisons of edge detection results on representative regions using four models: UEAD [56], PiDiNet [31], MuGE [55], and REAUNet-Sober [22]. From the visualizations, we observe that UEAD and MuGE tend to produce wider and blurred boundaries, indicating over-smoothed predictions that reduce localization precision. This issue is especially pronounced in densely terraced landscapes, where precise boundary localization is critical. In contrast, PiDiNet and REAUNet-Sober generate sharper and more compact edges, with REAUNet-Sober showing the best alignment with ground truth boundaries across complex spatial structures.

## E.3   Discussions the results for different regions for terraced parcel extraction

Fig. 16 compares three parcel-extraction networks—SEANet, REAUNet, and HBGNet—over eight performance metrics (Precision, Recall, IoU, Overall Accuracy, F1-Score, GOC, GUC, and GTC) across eight geographic zones. Overall, performance variance is most pronounced in Southwest and Other International Regions, underscoring the challenge posed by highly heterogeneous terrace morphology.

## E.4   Visualization cases on domain adaptation

Fig. 17 illustrate the segmentation results of four UDA methods (FDA, DAFormer, HRDA, and PiPa) on six representative domain transfer tasks, with transfers targeting the **North** (N), **South** (S), and

**Global** (G) domains, respectively. The results show that HRDA and PiPa generally produce finer boundaries and better preserve parcel structures under domain shift.

## F   Limitations and future work

Although GTPBD encompasses major terraced regions worldwide, the current collection is limited to fourteen well-known terraced areas in countries where terracing is concentrated. Some atypical terraced regions in other countries are not included due to their sparse distribution. The dataset currently consists only of optical images and does not include other remote sensing modalities such as multispectral or infrared imagery.

From the perspective of model training, GTPBD also presents several limitations. First, terraced parcels are highly fragmented and vary significantly in scale, which leads to class imbalance and ambiguous boundaries that pose challenges to current segmentation models. Second, although annotations were carefully generated, large-scale manual delineation inevitably introduces labeling noise, particularly in occluded or irregular regions. Third, the dataset mainly consists of single-season optical imagery, which limits its ability to capture temporal diversity; models trained under such conditions may underperform when applied to multi-season or multi-regional scenarios. Finally, while the resolution of 0.5–0.7 m is generally sufficient, extremely narrow ridges or complex land-cover mosaics remain difficult to capture and segment accurately.

In future work, we will expand the dataset by incorporating additional atypical terrace samples and broadening global coverage. Furthermore, GTPBD will be supplemented with multimodal geographical data, including Digital Elevation Models (DEMs), slope gradients, and spatiotemporal sequences. We also plan to construct question–answer pairs to facilitate integration with multimodal large language models. Finally, since the dataset already covers the primary terrace types, future tasks may focus on domain generalization and adaptation, enabling models not only to delineate known terraces but also to discover and adapt to new or atypical terrace types.

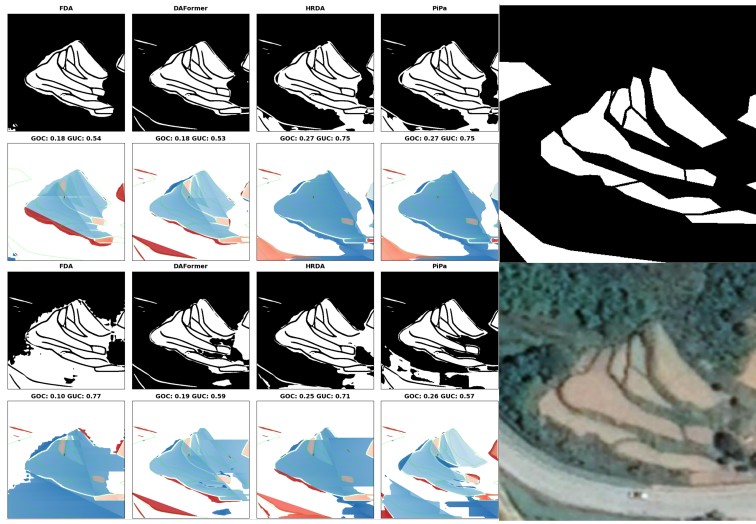

(a) UDA transfer to **North** (G→N top, S→N bottom).

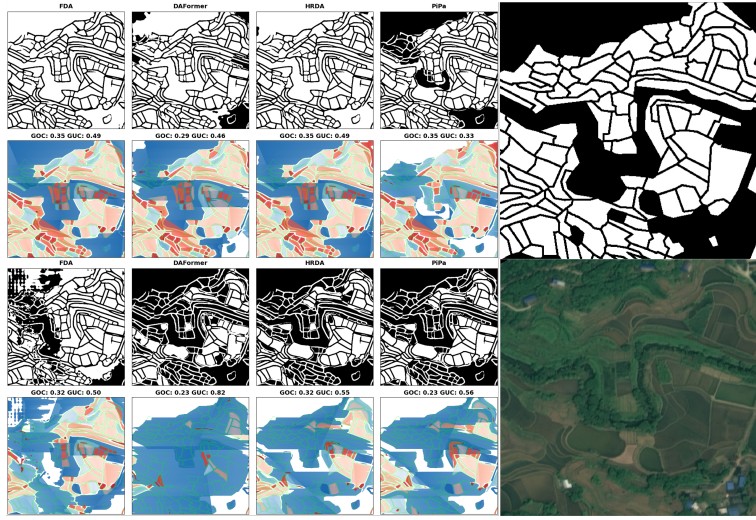

(b) UDA transfer to **South** (G→S top, N→S bottom).

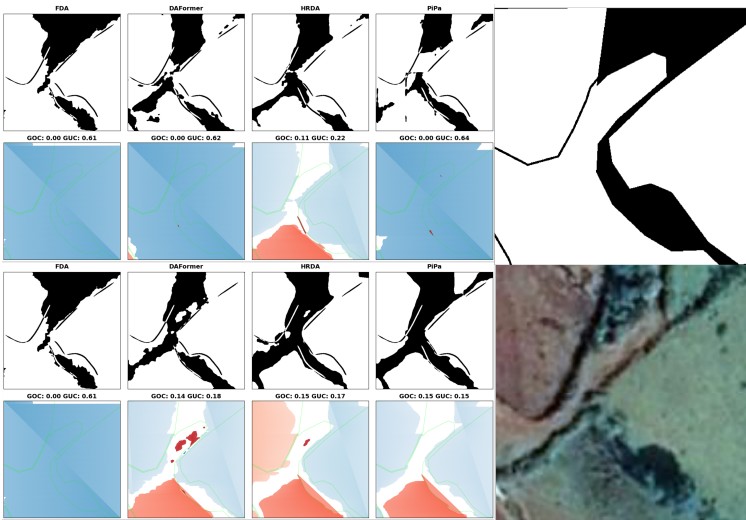

(c) UDA transfer to **Global** (S→G top, N→G bottom).

Figure 17: Qualitative results of UDA transfer to three domains: North, South, and Global. Models compared: FDA, DAFormer, HRDA, and PiPa.

