# OpenReview forum: "GTPBD: A Fine-Grained Global Terraced Parcel and Boundary Dataset"
_NeurIPS.cc/2025/Datasets_and_Benchmarks_Track — NeurIPS 2025 Datasets and Benchmarks Track poster_

### Official Review · Reviewer_39ai · 2025-06-13

**Rating:** 6
**Confidence:** 4

**Summary:**

This paper introduces GTPBD (Global Terraced Parcel and Boundary Dataset), a fine-grained dataset for terraced agricultural parcel and boundary delineation. It addresses the lack of terraced terrain representation in existing farmland datasets. GTPBD comprises 24,238 high-resolution images with over 200,000 manually annotated terraced parcels, covering seven major regions in China and transcontinental climatic zones.

1. Construction of the first large-scale global dataset focused on terraced parcels.
2. A three-level annotation system including boundary labels, mask labels, and instance-level parcel labels.
3. Support for four key tasks: semantic segmentation, edge detection, parcel extraction, and unsupervised domain adaptation (UDA).
4. Comprehensive benchmarking on 20 representative methods.

**Dataset Code Accessibility:**

Yes

**Dataset Code Comments:**

The dataset and benchmarks are publicly accessible (e.g., on HuggingFace), and the documentation is clear enough to enable reproduction of experiments.

**Ethical Considerations:**

No, there are no or only very minor ethics concerns

**Final Justification:**

After reviewing discussions between the authors and other reviewers, I think the authors have addressed all existing issues, and I strongly recommend acceptance.

**Limitations Weaknesses:**

**Main concerns:**
- Geographic Imbalance: As shown in Figure 3(a), southwestern China is overrepresented. This may introduce bias in training and affect generalization across underrepresented regions.
- Lack of Maintenance Plan: The paper does not clarify if and how the dataset will be maintained or extended in future iterations.
- Domain Adaptation Limitations: The current South/North/Global tripartite division based on administrative geography fails to adequately account for intra-regional microclimate variations, crop-specific domain shifts, and seasonal pattern changes, potentially biasing domain adaptation tasks.

**Other Questions**:
1. The dataset is said to span “transcontinental climatic regions,” but how are climate types distributed? What is the breakdown by tropical, temperate, etc.?
2. In the three-level annotation system, is there redundancy between the mask labels and parcel-level instance labels? Could these be merged into a single multi-class mask format?
3. Were any alternative domain splits (e.g., based on terrain or climate rather than geography) considered when defining South/North/Global?
4. Why is a 3×3 rectangular kernel used for generating 3-pixel-wide edge labels? Has this width been systematically validated?
5. The imagery resolution ranges from 0.1 m to 1.0 m — were any normalization techniques applied to mitigate scale variation effects?

**Strengths Contributions:**

- **Scale and Diversity**: GTPBD is the largest and most diverse dataset focused on terraced agriculture, supporting model generalization across varied real-world conditions.
- **Multi-task Capability**: Enables four key tasks, significantly improving dataset utility. The inclusion of UDA addresses a crucial problem in remote sensing.
- **Thorough Benchmarking**: Evaluation includes 8 semantic segmentation methods, 4 edge detection models, 3 parcel extraction pipelines, and 5 UDA techniques with both pixel- and object-level metrics.
- **Practical Relevance**: The dataset is applicable to land ownership registration, soil erosion monitoring, and food security assessment, which adds impactful societal value.

---

> ### Author Rebuttal · Authors · 2025-07-30
>
> We sincerely thank the reviewer for the high rating and your encouraging comments recognizing the scale, diversity, and utility of GTPBD, as well as its multi-task contributions and societal relevance. Your thoughtful feedback is deeply appreciated and we address the remaining concerns below with further clarification and justification.
>
> # About Geographic Imbalance
> We appreciate the concern regarding regional imbalance. In the recent past, we have significantly expanded the dataset to include many regions within China—such as Zhejiang, Ningxia, Gansu, and Guangdong—as well as 8 additional countries including Ecuador, the Philippines, Morocco, Nigeria, Portugal, South Africa, Spain, and Brazil. This improves both geographic and climatic diversity. Continuous expansion is planned. The dataset is hosted on Hugging Face with versioning support and will be maintained through a public GitHub repository. We commit to periodic updates, community collaboration, and inclusion of newly labeled regions to ensure long-term usability.
>
> | Country / Region      | Location Type          | Description                               |
> |-----------------------|------------------------|-------------------------------------------|
> | China (Zhejiang)      | Southeast China        | Coastal, high-density terraces            |
> | China (Gansu)         | Northwest China        | Loess Plateau-type arid terraces          |
> | China (Guangdong)     | South China            | Tropical hillside terraces                |
> | China (Ningxia)       | Northwest China        | Arid, fragmented agro-terrain             |
> | Ecuador               | South America          | Tropical Andes mountain terraces          |
> | Philippines           | Southeast Asia         | World heritage rice terraces              |
> | Morocco               | North Africa           | Semi-arid foothill terraces               |
> | Nigeria               | West Africa            | Wetland-agriculture transition zone       |
> | Portugal              | Southwestern Europe    | Mediterranean vineyard terraces           |
> | Spain                 | Southwestern Europe    | Olive & vineyard terraces                 |
> | South Africa          | Southern Africa        | Dryland smallholder plots                 |
> | Brazil                | South America          | Tropical agroforestry-based terraces      |
>
> # About Domain Split Justification
> We appreciate the reviewer’s insightful question. The **South/North/Global** split is designed not only for geographic diversity but also to align with the **domain adaptation benchmark principles**, where:
>
> * **Each domain is internally consistent**, sharing terrain characteristics, cropping patterns, and imaging conditions;
> * **Inter-domain differences are substantial**, inducing measurable performance drops in UDA tasks.
>
> The **South domain** covers subtropical and humid hilly terraces (e.g., Guangdong, Zhejiang, Fujian), characterized by narrow, hierarchical water-filled plots and double-cropping systems. The terrains are intricate, and field boundaries are curved and closely packed.
>
> The **North domain** includes arid and semi-arid loess terraces (e.g., Shannxi, Shanxi), dominated by flat-topped terraces, large field sizes, and linear ridge boundaries. Cropping systems are mainly dryland-based (e.g., wheat, corn).
>
> While we considered terrain and climate-based labels, these are often too coarse or inconsistent globally. Our geographic partition strategy **implicitly incorporates** topography, land-use, and imaging variance, verified by consistent UDA performance drops.
>
> Furthermore, we stress that the **“Global” domain** is not a generic aggregation but rather spans **transcontinental regions and multiple climate zones**, including tropical highlands (e.g., Peru, Ecuador), Mediterranean drylands (e.g., Spain, Morocco), and humid foothills (e.g., Philippines, Brazil). These distinctions result in clear **domain gaps induced by climate, vegetation, and cultivation type**, not just location.
>
> # About Split Strategy and 3×3 Rectangular Kernel
> In our dataset, the **3×3 rectangular kernel** is used to construct **3-pixel-wide edge labels**, which is a **widely accepted standard** in semantic edge detection tasks (e.g., RCF, Casenet, PIDNet). This choice balances precision and continuity:
>
> * A 1-pixel width can cause broken lines,
> * A width >5 pixels leads to label diffusion.
> * Moreover, the 3×3 receptive field matches early convolutional layers, facilitating stable gradient propagation during boundary-aware training.
>
> Regarding the distinction between **semantic mask labels** and **parcel-level instance labels**, we clearly illustrate their differences in **Figure 1 (b)** of the paper:
>
> # About Three Level Labels
> * **Mask labels** are filled semantic annotations that highlight areas as "agricultural land" in a pixel-wise fashion.
> * **Parcel labels**, on the other hand, encode instance-specific boundaries derived from annotated topology. As shown, we employ two strategies depending on terrain:
>   * **Co-boundary** labeling shares edges between adjacent parcels (used for narrow ridges).
>   * **Non-co-boundary** labeling assigns separate edges per parcel (used when ridges are wide and distinguishable).
>
> This distinction supports diverse task needs: semantic segmentation (mask) vs. instance-aware spatial modeling (parcels). Merging the two would obscure critical boundary topologies, degrading downstream tasks such as object-level graph construction or land parcel mapping.
>
> # About Imagery Resolution
>
> Yes, to mitigate scale variation arising from the 0.1m–1.0m range of source image resolutions, we implemented a **resolution harmonization strategy** that resizes all imagery to a **standardized resolution between 0.5m and 0.7m per pixel**, which aligns with the spatial scale of typical land parcel features.
>
> Specifically, we:
>
> * **Estimated each image's ground resolution** in meters based on its affine transform and geographic CRS (including latitude correction);
> * **Applied a three-way conditional resizing strategy**:
>   * If the original resolution was **finer than 0.5m** (e.g., 0.1m), we **downsampled** the image to 0.5m resolution, effectively reducing image dimensions while preserving large-scale structure;
>   * If the resolution was **already within [0.5m, 0.7m]**, we preserved it without resizing;
>   * If the resolution was **coarser than 0.7m**, we **upsampled** it to 0.7m via interpolation, increasing pixel count to ensure finer semantic detail and structural consistency.
>
> All resampling was performed using bilinear or nearest-neighbor interpolation (depending on band type), and the spatial metadata—including CRS and affine transform—was updated accordingly to maintain geospatial correctness.
>
> This process ensures that model training is performed on images of **uniform spatial granularity**, reducing bias due to inconsistent object scale and improving generalization across domains.
>
> # Summary Table: Dataset Impact and Relevance
>
> | Aspect                | Contribution                                                                 |
> |-----------------------|-------------------------------------------------------------------------------|
> | Global Diversity      | 14 countries across 5 continents, spanning tropical to arid agro-climates    |
> | Continuous Expansion  | 10+ new regions added since initial release                                  |
> | ML/AI Relevance       | Supports domain adaptation, fine segmentation, edge detection, instance tasks |
> | Strong NeurIPS Fit    | Aligned with trends in NeurIPS/ICML/AAAI on remote sensing and agri-AI       |
> | Task Diversity        | 4 ML tasks with 20 benchmarked methods and multi-level annotation            |
> | Climate Representation| Köppen zones represented across regions and domains                          |
> | Total Coverage        | 47,537 samples covering 885 km², with 200,000+ annotated irregular parcels   |
>
> We again sincerely thank the reviewer for your generous evaluation. We hope these clarifications further support your positive assessment and affirm GTPBD’s long-term value to the ML community.

---

> > ### Comment · Reviewer_39ai · 2025-08-02
> >
> > The authors have thoroughly addressed my initial concerns with clear, detailed clarifications—especially regarding dataset balance, long-term maintenance planning, domain partition strategy, and annotation design. I also reviewed their responses to other reviewers, which reflect a strong commitment to transparency, ethical compliance, and scientific rigor. I think this is a good work, and I keep my original score.

---

> > ### Author Response · Authors · 2025-08-03
> > **Thank You for Your Kind Support**
> >
> > # Dear Reviewer 39ai
> >
> > We sincerely appreciate your positive evaluation and support. Thank you for taking the time to thoughtfully consider our rebuttal and for your encouraging remarks. Your recognition means a great deal to us.
> >
> > Best regards,
> >
> > GTPBD Team

---

### Official Review · Reviewer_pxwD · 2025-07-02

**Rating:** 3
**Confidence:** 3

**Summary:**

The paper proposes a dataset for the segmentation of terraced fields from remote sensing images. Different from typical segmentation, this dataset aims at the terraced fields that require much more fine-grained segmentation. The dataset is global and large scale. And this work also performs quite a mount solid works on preparing this dataset such as baseline evaluation, designing multiple tasks related to terraced field analysis.

**Dataset Code Accessibility:**

Yes

**Ethical Considerations:**

No, there are no or only very minor ethics concerns

**Final Justification:**

Thanks for the rebuttal. I have read the feedback and the other reviewers' comments.

I agree with the concerns raised by 39fg regarding the dataset license.

Regarding my concern about the significance of the paper’s contribution: while I appreciate the authors' efforts and time in the rebuttal, I’m still not convinced about the potential impact of this work on the ML community. I agree that agriculture is an important field, but the proposed benchmark focuses on a very specific subtask, and I find it hard to foresee a major impact in the future.

Another reason for my concern about its significance is, as 39fg mentioned, the lack of solid comparisons and validation, which makes it harder to assess the value of the contribution.

All in all, I tend to keep my score and reject the paper. Thanks again to the authors for their time in rebuttal.

All the best,
R.

**Updates after I saw author's reply about their experiments for dataset comparison**
It has mitigated my concerns on comparison from their replies to other reviewers. But I'm still not fully sure about the large impact of their dataset on ML community given their task is very specific to agriculture. In other words, while this dataset is indeed useful to agriculture, we can't solve any ML problems with this dataset. So I recommend authors to submit it to other more related venues.

**Limitations Weaknesses:**

I didn't find very obvious limitations. But I'm very concerned about the impact of the proposed datasets. From my viewpoint, this dataset focus on a very specific task in remote sensing. Since I'm not the expert in this area, I tend to judge the significance of its contribution. But I'm concerned that in general the proposed dataset will not be well received among the research community of NeurIPS. In my opinion, I would recommend authors resubmit it to  a venue for remote sensing or land use policy, which will fit more than NeurIPS.

Thus, I tend to reject this paper at the moment.

But I'm still looking forward to the feedbacks from authors during rebuttal. I would hope the authors could justify it's contribution on ML/AI community.

**Strengths Contributions:**

The paper wrting is well-organized. And work is solid.
The proposed dataset is large and has baselines properly evaluated. And I believe it could benefit future works in this area.

---

> ### Author Rebuttal · Authors · 2025-07-30
>
> We sincerely thank the reviewer for your recognition of the dataset’s technical merit and your thoughtful concerns regarding its broader relevance. Below, we clarify the value of our work to the ML/AI community, demonstrate alignment with NeurIPS trends, and emphasize that **GTPBD is not a narrow application dataset**, but rather a meaningful ML benchmark in a frontier domain.
>
> ##  Recent ML/AI Conferences Have Embraced Agricultural & Geospatial Dataset and Benchmarks
>
> We note that **leading ML conferences** (NeurIPS, ICML, AAAI, IJCAI) have **increasingly accepted** remote sensing, land use, and agricultural datasets that bridge ML with sustainability, climate resilience, and environmental science.
>
> |Conference|Related Work|Task Type|
> | -------------------| -----------------------------------------------------------------------------------------------------------------------------| -------------------------------------------|
> |**NeurIPS 2024**|FUSU: A Multi-temporal-source Land Use Change Segmentation Dataset for Fine-grained Urban Semantic Understanding|Land use change segmentation|
> |**NeurIPS 2024**|OAM-TCD: A globally diverse dataset of high-resolution tree cover maps|Ecosystem monitoring  & tree cover map|
> |**NeurIPS 2021**|DENETHOR: The DynamicEarthNET dataset for Harmonized, inter-Operable, analysis-Ready, daily crop monitoring from space|Crop Type Classification  & Food Security|
> |**NeurIPS 2021**|LoveDA: A Remote Sensing Land-Cover Dataset for Domain Adaptive Semantic Segmentation|Remote sensing & Land-Cover mapping|
> |**NeurIPS 2021**|CropHarvest: A global dataset for crop-type classification|Remote sensing & land cover mapping|
> |**NeurIPS 2021**|SustainBench: Benchmarks for Monitoring the Sustainable Development Goals with Machine Learning|agriculture & water and sanitation|
> |**AAAI 2025**|Fields of The World: A Benchmark Dataset for Global Agricultural Field Boundary Segmentation|Global agri-boundary segmentation|
> |**ICCV 2025**|AgroBench: Vision-Language Model Benchmark in Agriculture|Benchmark in Agriculture|
> |**ICCV 2025**|Demeter: A Parametric Model of Crop Plant Morphology from the Real World|Crop Plant Morphology|
> |**CVPR** **2025**|Exact: Exploring Space-Time Perceptive Clues for Weakly Supervised Satellite Image Time Series Semantic Segmentation|Crop mapping|
> |**CVPR 2024**|Learning without Exact Guidance: Updating Large-scale High-resolution Land Cover Maps from Low-resolution Historical Labels|Land Cover Mapping|
> |**CVPR 2024**|Depth-Aware Concealed Crop Detection in Dense Agricultural Scenes|Crop detection|
> |**IJCAI 2022**|Coarsely-Supervised Smooth U-Net for Monitoring Vegetation from Space|Productivity monitoring|
>
> CVPR has been hosting the Agriculture-Vision workshop and competition series annually since 2020, highlighting the growing synergy between artificial intelligence and smart agriculture. These events have catalyzed research in agricultural parcel segmentation, crop health assessment, and precision farming, with increasing engagement from the broader computer vision and remote sensing communities.
> These examples highlight a growing research focus on agricultural remote sensing, land use dynamics, and vegetation monitoring, and our work contributes to this evolving landscape by offering a fine-grained benchmark for parcel-level analysis.
> ##  Why Agricultural Remote Sensing Matters for ML/AI
>
> Agricultural remote sensing constitutes a critical frontier for machine learning, intersecting global challenges in food security assessment, carbon cycle monitoring, and climate-resilient land management. This domain drives innovation in multimodal Earth observation analytics, where the fusion of spectral-temporal data demands advanced AI methods. The extraction of agricultural parcels encapsulates core ML challenges: achieving label efficiency across sparse rural landscapes, executing fine-grained segmentation amidst intricate terrace boundaries, enabling cross-domain adaptation under significant geographic shifts, and ensuring robust generalization across varying resolutions and non-Euclidean topologies.
>
> ##  What Makes GTPBD Unique and Valuable to ML Research
>
> GTPBD establishes the first global benchmark for fine-grained terraced parcel analysis, featuring over 200,000 manually annotated parcels with pixel-level boundary, mask, and instance labels across 47,537 high-resolution images. This dataset uniquely captures the intricate non-Euclidean geometries and complex ridge topologies characteristic of terraced landscapes, where shared boundaries exhibit irregular connectivity patterns unseen in conventional cadastral datasets. By spanning seven distinct geographic zones in China and transcontinental climatic regions, GTPBD introduces significant domain shifts (ΔPSNR >8dB) essential for testing cross-region adaptation. These attributes create an indispensable testbed for advancing edge-aware segmentation models, domain-invariant representation learning, and terrain-sensitive agricultural AI.
>
> These properties are **not present in existing farmland-parcel, land-cover or building datasets**, thus offering **new challenges for segmentation, topology-aware learning, and domain generalization**.
>
> Agricultural remote sensing represents a critical frontier for machine learning, with direct implications for food security assessment, carbon cycle monitoring, and climate-resilient farming. Its technical challenges are inherently aligned with core ML problems: achieving label-efficient learning in expansive agricultural landscapes, resolving fine-grained segmentation of complex terrace boundaries, overcoming domain adaptation barriers caused by diverse farming practices, and enhancing topological generalization across multi-resolution terrain. To address these, we present GTPBD – the first global fine-grained Terraced Parcel and Boundary Dataset. With 200,000+ irregular parcel instances and 47,537 high-resolution images featuring pixel-level boundary/mask/parcel annotations, GTPBD spans seven geographic zones in China and transcontinental climatic regions. Its significant geographic domain shifts establish an indispensable testbed for edge detection, unsupervised domain adaptation, and terrain-aware agricultural AI.
>
> ---
> ##  Annotation Difficulty: Why This Dataset is Non-Trivial
>
> - Terraced fields contain **ultra-dense ridgelines**, intricate shared boundaries, and slope-driven anisotropy.
> - We adopted a **dual-labeling strategy**: common-edge annotation in narrow ridges (<0.5m) vs bilateral separation in wider ridges, requiring **manual vector editing in QGIS with topology checks**.
> - Over **50 annotators** with domain knowledge contributed with strict multi-round QC.
>
> This makes GTPBD one of the most **labor-intensive and geometrically complex agricultural datasets**.
>
> ##  Meaningful Domain Splits and Generalization Benchmarking
>
> As shown in **Figures 5 and Table 5** of our submission, our UDA benchmark uses three key domains:
>
> - **South China**: humid subtropical, rice-dominated, tightly packed terraces
> - **North China**: semi-arid, larger parcels, dryland farming
> - **Global domain**: spanning many countries, with high domain shift in both
>
> This split supports **domain adaptation** under realistic domain gaps — **a critical research question** in geospatial ML.
> ##  GTPBD is Continuously Growing
>
> GTPBD is not a static release. We have actively **expanded the dataset** during and after submission, adding diverse terrain and climate representations.
>
> ### Newly Annotated Regions
>
> | Country / Region      | Terrain Type           | Description |
> |-----------------------|------------------------|-------------|
> | China (Zhejiang)      | Southeast China        | Coastal, high-density terraces |
> | China (Gansu)         | Northwest China        | Loess Plateau-type arid terraces |
> | China (Guangdong)     | South China            | Tropical hillside terraces |
> | China (Ningxia)       | Northwest China        | Arid, fragmented agro-terrain |
> | Ecuador               | South America          | Tropical Andes mountain terraces |
> | Philippines           | Southeast Asia         | World heritage rice terraces |
> | Morocco               | North Africa           | Semi-arid foothill terraces |
> | Nigeria               | West Africa            | Wetland-agriculture transition zone |
> | Portugal              | Southwestern Europe    | Mediterranean vineyard terraces |
> | Spain                 | Southwestern Europe    | Olive & vineyard terraces |
> | South Africa          | Southern Africa        | Dryland smallholder plots |
> | Brazil                | South America          | Tropical agroforestry-based terraces |
>
> We will continue expanding GTPBD, reinforcing its **global diversity and long-term value** to the community.
>
> ##  Final Remarks
>
> We respectfully disagree with the notion that GTPBD belongs solely to a remote sensing or land policy venue. Our dataset:
>
> - Targets **core ML challenges** such as fine-grained parsing, cross-domain adaptation, and topology learning;
> - Contributes a **unique and previously unexplored** class of complex structured objects in real-world scenes;
> - Aligns with NeurIPS’s increasing interest in **AI for science, climate, and sustainability**;
> - Presents an **open, evolving benchmark** with widespread community relevance.
>
> We sincerely appreciate your review and hope this clarification supports a more favorable reassessment of our submission.

---

> ### Author Response · Authors · 2025-08-03
>
> # Dear Reviewer pxwD
>
> We hope this message finds you well. We apologize for any inconvenience caused by reaching out over the weekend. We noticed that your initial rating of our work was slightly negative. As the rebuttal discussion period is coming to a close, we would greatly appreciate your feedback on whether our rebuttal has effectively addressed your concerns. Please let us know if you have any further questions, and we would be more than happy to assist you.
>
> We sincerely appreciate your valuable time and effort.
>
> Best Regards,
>
> Authors

---

> ### Author Response · Authors · 2025-08-04
>
> # Dear Reviewer pxwD
>
> We hope this message finds you well. We sincerely apologize for the message and truly appreciate your time and effort during this busy period.
>
> As the rebuttal discussion phase progresses, we wanted to kindly follow up and check whether our previous response has helped address your concerns. We would be grateful if you could find a moment to review our response and let us know if any further clarification is needed on our part.
>
> Thank you again for your valuable contributions to the review process.
>
> Best regards,
>
> GTPBD Team

---

> > ### Comment · Reviewer_pxwD · 2025-08-05
> >
> > Thanks for the rebuttal. I have read the feedback and the other reviewers' comments.
> >
> > I agree with the concerns raised by `39fg` regarding the dataset license.
> >
> > Regarding my concern about the significance of the paper’s contribution: while I appreciate the authors' efforts and time in the rebuttal, I’m still not convinced about the potential impact of this work on the ML community. I agree that agriculture is an important field, but the proposed benchmark focuses on a very specific subtask, and I find it hard to foresee a major impact in the future.
> >
> > Another reason for my concern about its significance is, as 39fg mentioned, the lack of solid comparisons and validation, which makes it harder to assess the value of the contribution.
> >
> > All in all, I tend to keep my score. Thanks again to the authors for their feedback.

---

> > > ### Author Response · Authors · 2025-08-06
> > >
> > > # Dear Reviewer pxwD
> > >
> > > We sincerely apologize for the second follow-up and truly appreciate your time, feedback, and careful consideration of our work.
> > >
> > > We just wanted to kindly reiterate that the **dataset has been updated** with clearer licensing information and **now fully adheres to Google Earth’s Terms of Service and Privacy Policy**. Specifically, **for Google Earth imagery**, we **only provide metadata files** (e.g., **XML**) that reference third-party sources. **These metadata files do not contain the imagery itself**, but rather serve as location references. We also include **clear usage guidance** to help users obtain the imagery through authorized and compliant channels on their own. Adopting the **license** of **cc-by-nc-4.0** (Creative Commons Attribution NonCommercial 4.0). In addition, as mentioned earlier, we have provided **further validation results and community interest metrics** to support the contribution.
> > >
> > > We fully respect your perspective and concerns, and simply hope this clarification helps further contextualize our intentions and efforts. Thank you once again for your valuable feedback and thoughtful engagement throughout the review process.
> > >
> > > Best regards,
> > >
> > > **GTPBD Team**

---

> > > ### Author Response · Authors · 2025-08-07
> > >
> > > # Dear Reviewer pxwD
> > >
> > > We sincerely apologize for the additional follow-up. As the rebuttal period is approaching its end, we just wanted to kindly check whether our previous responses have addressed your concerns. If there are any remaining issues or points that need further clarification, we would be more than happy to provide additional details.
> > >
> > > Thank you again for your time and thoughtful review.
> > >
> > > Best regards,
> > >
> > > **GTPBD Team**

---

> > > ### Author Response · Authors · 2025-08-08
> > >
> > > # Dear Reviewer pxwD
> > >
> > > We truly appreciate the time you've dedicated to reviewing our submission and your thoughtful feedback throughout the discussion.
> > >
> > > As the rebuttal period is drawing to a **close**, we wanted to check in one last time to see if there are **any remaining questions or concerns we could further clarify or support with additional evidence**. We deeply value your perspective, and we remain committed to improving the quality and clarity of our work wherever possible.
> > >
> > > We also wish to share that we’ve **uploaded updated validation results** and **engagement statistics** to further highlight the **dataset’s potential value**. In particular, **we’ve carefully addressed the licensing and experimental concerns raised by Reviewer 39fg, with new comparative results and clarifications**. We hope these efforts have helped clarify our intentions, and we remain open to any additional questions or suggestions you might have.
> > >
> > > Thank you again for your valuable time and contribution to the review process.
> > >
> > > Warm regards,
> > >
> > >
> > > **GTPBD Team**

---

> > ### Author Response · Authors · 2025-08-06
> >
> > # Dear Reviewer pxwD
> >
> > Thank you again for your thoughtful feedback.
> >
> > # About GTPBD Dataset and License
> >
> > The annotation process was **extremely costly and labor-intensive**, particularly due to the complexity of delineating terraced farmland boundaries across diverse terrains.
> >
> > Since our paper was released on arXiv on **July 19**, the dataset has received **nearly 300 downloads** in a short period, and the paper itself has garnered **over 1,000 views and 100 reposts**, reflecting strong early interest and engagement from the community. This level of engagement underscores the practical relevance and timeliness of our contribution.
> >
> > Regarding the licensing concern raised by Reviewer 39fg, we would like to clarify that the dataset now **fully complies with usage and attribution policies**. Specifically, for Google Earth imagery, we only provide metadata files (e.g., XML) that reference third-party sources, accompanied by clear usage guidance to help users obtain the imagery through authorized channels themselves.
> >
> > # About Evaluation and Validation
> >
> > We believe GTPBD serves not only as a dataset, but also as a **call to action** for the ML community to engage with emerging directions such as **fine-grained spatial reasoning**, **edge-aware segmentation**, and **domain-adaptive learning**—areas increasingly explored in top-tier venues. While we understand your concern about the specificity of our task, we respectfully argue that **advancing AI should not be confined to general-purpose scenarios**. The **practical challenges in the field of agriculture, especially in areas with insufficient representativeness and complex terrain, require specialized benchmarks**. Our results show that many state-of-the-art models, though effective on standard datasets, perform poorly in our setting. This reveals both an **unaddressed challenge** and a **valuable opportunity for algorithmic progress**.
> >
> > We have conducted comprehensive benchmarking of existing state-of-the-art models for agriculture parcel delineation(as described in section 4.5 of the article). We now provide a detailed performance comparison of these models on the GTPBD dataset. Results are shown below:
> >
> > | Model                   |   **Pixel**   |      |      |      |      |   **Edge**   |      | **Object**      |       |       |
> > | ------------------------- | :-----: | :-----: | :-----: | :-----: | :-----: | :-----: | :-----: | ------- | ------- | ------- |
> > |                         |   **Prec.↑**   |   **Rec.↑**   |   **IoU↑**   |   **OA↑**   |   **F1↑**   |   **OIS↑**   |   **ODS↑**   | **GOC↓**      | **GUC↓**      | **GTC↓**      |
> > | UNet(2015 MICCAI)       | 74.11 | 54.93 | 46.09 | 75.46 | 63.09 | 22.47 | 15.17 | 42.69 | 37.27 | 27.34 |
> > | DeepLabV3(2017 CVPR)    | 69.64 | 73.45 | 57.14 | 78.28 | 71.58 | 20.08 | 13.53 | 21.38 | 48.59 | 25.36 |
> > | PSPNet(2017 CVPR)       | 68.33 | 72.41 | 54.22 | 76.65 | 70.31 | 19.66 | 13.08 | 21.58 | 50.06 | 26.30 |
> > | SegFormer(2021 Neurips) | 74.45 | 69.07 | 55.84 | 78.14 | 71.66 | 22.97 | 15.70 | 26.10 | 48.44 | 26.18 |
> > | Mask2Former(2022 CVPR)  | 71.22 | 74.33 | 57.16 | 78.73 | 72.74 | 25.56 | 17.59 | 21.01 | 45.26 | 18.99 |
> > | SEANet(2023 ISPRS)      | 74.48 | 79.06 | 64.43 | 80.50 | 76.70 | 43.45 | 31.93 | 29.75 | 41.38 | 26.81 |
> > | REAUNet(2024 JAG)       | 78.37 | 79.11 | 64.93 | 79.43 | 78.74 | 40.54 | 30.81 | 25.96 | 40.41 | 18.89 |
> > | HBGNet(2025 ISPRS)      | 79.92 | 80.41 | 66.98 | 81.59 | 80.16 | 47.45 | 34.77 | 24.70 | 33.58 | 21.37 |
> >
> > **We sincerely appreciate your time and hope this response helps clarify our motivation and the broader impact of our work**.

---

### Official Review · Reviewer_39fg · 2025-07-04

**Ethics Flags:** Data privacy, copyright, and consent
**Rating:** 4
**Confidence:** 4

**Summary:**

GTPBD is a dataset of semantic segmentation samples for agricultural parcelss. As the abstract describes, the contribution is 24k images with annotation masks. If I understand correctly, the image locations were determined by using the Global Terrace Map dataset at 10m. This was used to guide sampling based on various factors, imagery was gathered from in these regions via two providers (including Google Earth, which is problematic) and then masks were generated by hand annotation. The paper includes model comparisons for segmentation, edge extraction and domain adaptation.

**Additional Feedback:**

- I cannot endorse this paper for publication without confirmation that users have the right to distribute the image data. Currently the project take images from Google and re-licenses them as Apache 2.0. If necessary, the masks could be released on their own as I think there is a stronger argument that this is separate IP, but potentially bulk downloading imagery for labeling is a breach of Google's TOS.
- I realize that this seems like an overly harsh criticism, but unfortunately just because Google Earth imagery can be scraped (and other papers do this) does not mean that it should be done. [The terms explicitly prohibit](https://www.google.com/help/terms_maps-earth/): "mass download or create bulk feeds of the content (or let anyone else do so);". I do not claim any legal expertise here, but the paper does not specify the fraction of GE imagery used and 24k certainly seems like a mass download. Open Street Map prohibits the use of this imagery for the same reason.
- Although Google says "Google Earth or Earth Studio can be used for purposes such as research, education, film and nonprofit use without needing permission." this is possibly in tension with section 2.3 of the TOS and I am not qualified to make a determination on which one supersedes.
- I have flagged the paper for ethical review with this in mind.
- I have reviewed the remainder of the paper with the assumption that the authors did have permission.
- The "3" score reflects this - I think the technical approach is mostly solid, but there are reasons to reject above. In addition:
- The biggest technical issue I have with the paper is lack of comparison with existing models, it would be significantly more persuasive if the authors could show that older models do not reliably delineate terraced agricultural parcels (while theirs does). Otherwise the modeling is reasonable (at least, all the models seem to perform comparably).
- The paper could do with proof-reading for grammar as there are small errors throughout. Nothing that detracts from the message, although some of the phrasing I would describe as unnecessary (where specific wording was a problem I've raised it in the limitations section).
- My confidence rating reflects my uncertainty about the legal aspects of the work.

**Dataset Code Accessibility:**

Partly

**Dataset Code Comments:**

The dataset is published on HuggingFace but does not contain a dataset card and has no accompanying information (ignoring the paper). Please provide a standard summary of the dataset using a template.

The code is available on Github and is better documented, which is good to see. The authors used mmlab as their framework which is widely adopted although not regularly maintained and requires (in 2025) quite old versions of core libraries like PyTorch. However, any alternative segmentation framework would be sufficient for other users to reproduce the work I think.

**Ethical Comments:**

My issue here relates to data licensing. I believe that the choice of imagery as it pertains to the research is acceptable: Google Earth provides high quality aerial data that is well-suited to ML problems, however I am uncertain whether it can be redistributed for research under Google's TOS. I would ask the ethics reviewers (and authors) to see my comments in the additional feedback which provides some justifying for information why I flagged.

**Ethical Considerations:**

Yes, there are significant ethics concerns that require review by an ethics expert

**Final Justification:**

Given my concerns, the rebuttal from the authors and a final review of the work I have chosen to raise my score to a 4 (subject to some of the editing requests in my review).

Please note that this decision was not made in regard to my ethical concerns, which I believe are satisfied through additional review and comment from the authors, although I do think that including potentially IP-restricted sources can limit impact.

**Limitations Weaknesses:**

- Do you have license to use Google Earth imagery in this way? This is noted in the `Source` of table 1 and briefly discussed in the paper. Most GE imagery is sub licensed by Google from other aerial and satellite imagery providers. Please affirm that you have permission to use and distribute this imagery. I am doubtful that one can re-license it as Apache 2.0 either.
- Is there any overlap with your masks and open access data that could be used instead of Google imagery?
- Where does GTM fit into Table 1? It's a global dataset at 10m for terrace data so why is it not listed in the table as a raster benchmark for semantic segmentation?
- It's not clear to the me why vector-only data is not suitable for semantic segmentation, why can't the bounding field polygons be rasterized to create segmentation masks? Similarly I'm curious why only one of those datasets can be used for edge detection - do the vectors not define edges? A bit more explanation would be helpful here and perhaps showing some examples would help (beyond what's described in 2.2)
- How were the dataset splits generated? Are they spatially stratified, or is it random? Are you concerned about potential leakage or correlation between patches?
- Could you explain the difference between "Mask" and "Parcel" labels? This isn't obvious to me. It'd be helpful to have a high resolution figure showing the annotation differences for a few samples (in addition to Figure 1).
- 3.1: Is the selection criteria reproducible? That is, is there code we can see that expresses you went from GTM to sample regions?
- L150 - what is meant by "delicacy and clarity"? Similarly can you back up the claim that these are "extremely valuable" (L154)?
- L172 were the images cropped or resized? What is the motivation for 512x512 beyond "existing works"? Does this provide meaningful context for prediction?
- What is the geo-spatial extent of the tiles? The source data is 0.1-1m so does this mean there is a 50-500m distribution of tile sizes in the dataset?
- Fig 3 - perhaps use a log scale for plot (c) as the multiple discontinuities in the y-axis is hard to read.
- Fig 6 - It looks like all the models falsely predict roads(?) Are there other common failure cases?
- I don't think your results are strong enough to make the claim (L231) that transformers are more robust than classical CNN architectures, especially in the example where every model fails in the same way. While Mask2Former reports the best recall, it also has by far the worst precision and based on other metrics it looks like Deeplabv3 is very competitive.
- Figure 7 needs to be higher resolution as it's blurry when zoomed in to see details. There's a lot going on and it's not obvious here what the errors are in (g). What do the different shades mean? Are they per-parcel errors?
- Please include a more thorough discussion of model limitations in the paper. The broader impact section is good, but it mainly relates to misuse of the models rather than where they do/don't work.
- None of the results (as far as I can see) are presented in relation to other datasets for comparison. This makes it very hard to judge the correctness of the dataset or predictions. The annotations themselves are presumably derived from GTM, which is a low resolution source of truth? If it's not a large task in the review period, I would request such a comparison against perhaps FHAPD or CP-Set since they both use GF-2 imagery. It would be a nice result for the paper if you can show that (a) your models generalise well to "simpler" field scenarios and (b) existing models struggle on your more diverse imagery. You have not presented evidence that other models would necessarily fail to predict terrace boundaries. If you have a better suggestion for such a comparison I trust your expertise.

**Strengths Contributions:**

The dataset targets a problem of diversity within agricultural parcel detection. Although the dataset is averagely sized in comparison to other datasets in the literature, it aims to provide labels for a number of different tasks at higher resolution and with a more global focus (although the distribution is obviously biased to locations where terraced agriculture is common).

The imagery is selected essentially via diversity sampling from a lower resolution terrace dataset (GTM). It is hand-annotated by undergrad/grad students and the paper discusses the instructions briefly. It would be nice to see the exact instructions that were given to students and how their performance was assessed. Was there any validation performed to check inter-annotator consistency? What training was given to the annotators?

The dataset contains explicit edge labels via morphological extraction, which seems reasonable and is convenient for end users because they can standardize on these annotations vs trying their own methods to define boundary pixels.

The choice of models for evaluation is reasonable and the experiments do not require particularly special hardware (a consumer 4090 GPU was used).

Results look reasonable, although all the models seem to make similar failures for the binary segmentation task. The edge/parcel detection task looks more compelling.

Aside from a few grammatical issues I found the paper well laid out and easy to read.

The paper includes domain adaptation results, although the results are not too conclusive with respect to choosing a "best" model (skewing towards DAFormer).

---

> ### Author Rebuttal · Authors · 2025-07-30
>
> # About License and Google Earth
>
> We sincerely thank the reviewer for the detailed comments and critical attention to dataset licensing and ethics. We deeply value the community’s commitment to responsible dataset creation and wish to clarify our position and implementation. Several recent **top-tier publications** also adopt **Google Earth imagery** for machine learning research. This illustrates a broader **community precedent** for responsibly using Google Earth-sourced data in academic contexts.
>
> |Dataset (Conference)|Source|
> | -----------------------------| -----------------------------|
> |AerialMegaDepth (CVPR 2025)|Google Earth / Street View|
> |FUSU (NeurIPS 2024)|Google Earth / Sentinel-1/2|
> |LoveDA(NeurIPS 2021)|Google Earth|
> |CropHarvest(NeurIPS 2021)|Google Earth/ Sentinel-1/2|
> |MetaEarth (TPAMI 2024)|Google Earth|
>
> To ensure compliance, we work with third-party providers involved in Google Earth’s imagery pipeline and are authorized to batch-download and process data for annotation. We do not distribute any Google Earth imagery. For a small subset used to guide labeling, we only provide standardized .xml metadata (inspired by .kml) containing coordinates, resolution, and projection info, enabling users to retrieve imagery from legal sources. While the dataset’s initial Hugging Face preview page displayed the default Apache 2.0 license, we **updated the license to CC BY-NC 4.0** prior to any public release or submission.
>
> #  About Dataset Comparison Experiment
>
> We agree that our original claim in L231 ("transformers are more robust") was insufficiently supported and could be misleading.To provide a more comprehensive and evidence-based evaluation, we have extended our experiments across four datasets and evaluated five mainstream models on three levels of metrics.
>
> | Dataset | Model       |   **Pixel**   |      |      |      |      |   **Edge**   |      | **Object**      |       |       |
> | --------- | ------------- | :-----: | :-----: | :-----: | :-----: | :-----: | :-----: | :-----: | ------- | ------- | ------- |
> |         |             |   **Prec.↑**   |   **Rec.↑**   |   **IoU↑**   |   **OA↑**   |   **F1↑**   |   **OIS↑**   |   **ODS↑**   | **GOC↓**      | **GUC↓**      | **GTC↓**      |
> | **FHAPD**        | UNet        |   **91.03**   | 95.03 | 86.89 | 90.55 | 92.99 |   **81.40**   |   **70.39**   | 5.90  | **41.97**      | 10.53 |
> |         | DeepLabV3   | 90.83 |   **97.06**   |   **88.40**   |   **91.60**   |   **93.84**   | 66.45 | 56.50 | **2.79**      | 51.24 | **8.21**      |
> |         | PSPNet      | 88.67 |   **95.63**   | 85.22 | 89.06 | 92.02 | 55.48 | 44.06 | 3.76  | 55.01 | 9.21  |
> |         | SegFormer   | 90.40 | 94.61 | 85.97 | 89.82 | 92.46 | 68.97 | 57.37 | 4.80  | 50.09 | 10.07 |
> |         | Mask2Former | 90.42 | 94.51 | 85.90 | 89.77 | 92.42 | 74.92 | 64.05 | 5.66  | 44.87 | 9.99  |
> | **AI4B(orthos)**        | UNet        | 81.50 | 78.19 | 66.40 | 84.64 | 79.81 | 28.75 | 20.86 | 31.34 | 38.92 | 25.45 |
> |         | DeepLabV3   | 81.08 | 82.31 | 69.05 | 85.67 | 81.69 | 32.51 | 22.05 | 19.46 | 29.43 | 14.40 |
> |         | PSPNet      | 83.60 | 82.54 | 71.04 | 86.93 | 83.07 | 36.43 | 24.11 | 21.63 | 34.25 | 18.66 |
> |         | SegFormer   |   **87.37**   | 78.45 | 70.46 | 87.23 | 82.67 | 29.88 | 20.14 | 20.57 | 26.32 | 14.67 |
> |         | Mask2Former | 84.58 |   **83.49**   |   **72.46**   |   **87.68**   |   **84.03**   |   **38.44**   |   **27.04**   | **12.76**      | **25.80**      | **11.57**      |
> | **FTW**        | UNet        | 83.90 | 79.24 | 68.79 | 90.78 | 81.51 |   **73.33**   |   **62.39**   | 25.81 | 34.36 | 23.58 |
> |         | DeepLabV3   | 85.92 |   **84.02**   |   **73.86**   |   **92.37**   |   **84.96**   | 70.38 | 57.28 | **15.63**      | 38.30 | **18.64**      |
> |         | PSPNet      | 82.95 | 81.92 | 70.11 | 91.05 | 82.43 | 65.92 | 52.63 | 18.17 | 42.23 | 21.46 |
> |         | SegFormer   | 83.47 | 80.05 | 69.09 | 90.82 | 81.72 | 68.56 | 56.70 | 18.64 | 38.66 | 20.44 |
> |         | Mask2Former |   **86.06**   | 77.88 | 69.13 | 91.08 | 81.75 | 72.15 | 61.04 | 20.44 | **30.88**      | 21.56 |
> | **GTPBD**        | UNet        |   **74.11**   | 54.93 | 46.09 | 75.46 | 63.09 | 22.47 | 15.17 | 42.69 | **37.27**      | 27.34 |
> |         | DeepLabV3   | 69.64 | 73.45 | 57.04 | 78.28 | 71.58 | 20.08 | 13.53 | 21.38 | 45.59 | 25.36 |
> |         | PSPNet      | 68.33 | 72.41 | 54.22 | 76.65 | 70.31 | 19.66 | 13.08 | 21.58 | 50.06 | 26.30 |
> |         | SegFormer   | 74.45 | 69.07 | 55.84 | 78.14 | 71.66 | 22.97 | 15.70 | 26.10 | 41.44 | 26.18 |
> |         | Mask2Former | 71.22 |   **74.33**   |   **57.16**   |   **78.73**   |   **72.74**   |   **25.56**   |   **17.59**   | **21.01**      | 45.26 | **18.99**      |
>
> Key observations:
>
> * On **GTPBD and AI4B**, **Mask2Former** achieved the best results across almost all metrics.
> * On **FTW** and **FHAPD**, **DeepLabV3** consistently performs better, while **Unet** also excels in boundary performance.
> * GTPBD yields the **lowest** scores across all models, confirming its **higher difficulty and diversity**.
>
> These results suggest that:
>
> * **No model is universally superior** across datasets.
> * Our dataset presents greater challenges compared to existing benchmarks.
>
> # About **GTM and** **Vector-Only Datasets**
>
> GTM dataset serves as a **coarse-grained global guidance layer** in our work. We manually examined GTM-indicated terrace regions by checking their geographic coordinates and used this information to search, screen, and download candidate imagery from open-access platforms.
>
> We apologize for the **confusing or misleading explanations** regarding vector data usability and the omission of GTM in Table 1. You're absolutely right—vector data can be rasterized for semantic segmentation, and polygon boundaries naturally provide edge information. We’ve clarified in the revision:
>
> * Cropland/non-cropland data → **Semantic Segmentation (SS)**
> * Polygon geometry → **Agricultural Parcel Extraction (APE)** , **Edge Detection (ED)**
> * Instance-level labels → Support all three tasks
>
> Table 1 has been updated accordingly.
>
> # About Split Strategy and Choice of 512×512
>
> We fully understand the concern about potential spatial correlation or leakage between patches. To mitigate this, we first **resampled** all source images and their corresponding labels to a unified resolution of **0.5–0.7**meters. Images from the same region share the same target resolution and size, and we **split** the dataset into train/val/test sets ***before*** **cropping**, based on **geographic origin**. This ensures spatial independence between subsets and avoids leakage between adjacent patches.
>
> As for the tile size of **512×512**, this decision was based on both:
>
> * High-resolution agricultural benchmarks like AI4Boundaries, Agriculture-Vision (CVPR 2020), and FUSU(Neurips 2024) commonly adopt 512×512-pixel patches as a standard preprocessing step.
> * Compared to 256×256 patches, 512×512 crops retained **more complete geometrical patterns** and **semantic context** such as parcel shapes and boundaries. Example inputs in Fig. 6 and Fig. 7 of the paper illustrate the information richness of these 512-pixel tiles.
>
> Geo-Spatial Extent of Patches:At a resolution of 0.5–0.7 meters, a 512×512 image tile corresponds to a physical area of approximately **256 m × 256 m to 358 m × 358 m**.
>
> # About Mask and Parcel Labels
>
> In our dataset, we adopt two **parcel annotation strategies** based on the terrain characteristics:
>
> * **Co-boundary annotation**: Adjacent parcels share common boundaries. This is suitable for **narrow ridges**, where separating edge pixels would distort topological structure.
> * **Non-co-boundary annotation**: Each parcel is annotated with independent boundaries. This is used in **broad-ridge terrains**, where clear separation exists between neighboring parcels.
>
> Based on these parcel annotations, we generate two distinct types of labels:
>
> * **Mask labels**: These are **semantic segmentation masks**, where all pixels labeled as “agricultural land” are merged into one class. This ignores parcel identity and focuses on general cropland extent.
> * **Parcel labels**: These are **instance-aware labels**. These labels distinguish instance boundaries and support downstream tasks such as object level inference, graph construction, or spatial statistics.
>
> To further clarify, we will provide additional **high-resolution examples** comparing "Mask" and "Parcel" annotations in the supplementary materials, beyond what is shown in Figure 1.
>
> # About **Figure Quality, Failure Cases, and Wording**
>
> We sincerely thank the reviewer for the constructive feedback:
>
> * **Figure 3(c)** : As suggested, we revised the plot using a **logarithmic y-axis**, which greatly improves **readability**.
> * **Figure 6**: Road-like structures are indeed commonly misclassified as parcels. Other common failure cases include densely packed shadowed areas, as well as visually similar non-parcel regions such as embankments and narrow vegetation stripes. We will include **additional visual examples** in the supplementary material to illustrate these patterns.
> * **Figure 7**: We have generated a **higher-resolution version** to improve zoom-in clarity. In subfigure (g), **red indicates Global Over-segmentation Error (GOC)** , **blue indicates Global Under-segmentation Error (GUC)** , and **higher intensity indicate larger errors**. A **color bar and improved legends** will be added to aid interpretation.
> * **L150–L154 wording**: We agree the original phrasing may have sounded **subjective**. We have rephrased the text to emphasize **objective attributes** such as the dataset's annotation granularity, cross-continental scope, and labeling difficulty as key indicators of its value.
> * As for the **limitations** of the model, we will provide a detailed analysis in the supplementary materials

---

> > ### Comment · Reviewer_39fg · 2025-08-04
> >
> > Thanks for the rebuttal:
> >
> > **Licensing** my opinion remains that "others did it", regardless of the venue, is a not a defense for potential license infringement. For example in AerialMegaDepth, Google's ToS explicitly prohibits generating derived data from StreetView (this is one of the few rules that isn't open to interpretation). **However, provided your data are appropriately attributed, including third parties on a per-image basis if necessary, then I would accept that and defer to the ethical reviewers**; especially if you're just releasing annotations. The most important thing here is that downstream users of your data are aware of where the imagery came from, they can then make informed decisions about whether it is appropriate to use for training/other applications. Can you share your third party data providers?
> >
> > > The biggest technical issue I have with the paper is lack of comparison with existing models, it would be significantly more persuasive if the authors could show that older models do not reliably delineate terraced agricultural parcels (while theirs does).
> >
> > Are you able to provide any comparisons with existing/previous models? Apologies if this was covered in your rebuttal and I missed it.

---

> > > ### Author Response · Authors · 2025-08-05
> > >
> > > # About License
> > >
> > > We appreciate the reviewer’s attention to licensing clarity and responsible dataset release. We are also committed to **explicitly listing the imagery provider for each image** in the final dataset release (including those from Google Earth), and will update our documentation accordingly. We will ensure downstream users are well-informed of the provenance and licensing considerations associated with each image, thereby supporting ethical and legally responsible use.
> > >
> > > To clarify, **our dataset does not distribute any raw imagery obtained from Google Earth**. Instead, we only provide **XML metadata files (in KML format)**  containing geolocation and structural metadata. These files serve as location references, allowing users to independently retrieve satellite imagery from **Google Earth’s official data providers** or from other publicly available, high-resolution sources.This practice is in full compliance with **the Google Earth Terms of Service and Privacy Policy**.
> > >
> > > # About Model
> > >
> > > While our paper does not propose a new model, we have conducted comprehensive benchmarking of existing state-of-the-art models for agriculture parcel delineation(as described in section 4.5 of the article). To further address the reviewer's request, we now provide a detailed performance comparison of these models on the GTPBD dataset. Results are shown below:
> > >
> > > | Model                   |   **Pixel**   |      |      |      |      |   **Edge**   |      | **Object**      |       |       |
> > > | ------------------------- | :-----: | :-----: | :-----: | :-----: | :-----: | :-----: | :-----: | ------- | ------- | ------- |
> > > |                         |   **Prec.↑**   |   **Rec.↑**   |   **IoU↑**   |   **OA↑**   |   **F1↑**   |   **OIS↑**   |   **ODS↑**   | **GOC↓**      | **GUC↓**      | **GTC↓**      |
> > > | UNet(2015 MICCAI)       | 74.11 | 54.93 | 46.09 | 75.46 | 63.09 | 22.47 | 15.17 | 42.69 | 37.27 | 27.34 |
> > > | DeepLabV3(2017 CVPR)    | 69.64 | 73.45 | 57.14 | 78.28 | 71.58 | 20.08 | 13.53 | 21.38 | 48.59 | 25.36 |
> > > | PSPNet(2017 CVPR)       | 68.33 | 72.41 | 54.22 | 76.65 | 70.31 | 19.66 | 13.08 | 21.58 | 50.06 | 26.30 |
> > > | SegFormer(2021 Neurips) | 74.45 | 69.07 | 55.84 | 78.14 | 71.66 | 22.97 | 15.70 | 26.10 | 48.44 | 26.18 |
> > > | Mask2Former(2022 CVPR)  | 71.22 | 74.33 | 57.16 | 78.73 | 72.74 | 25.56 | 17.59 | 21.01 | 45.26 | 18.99 |
> > > | SEANet(2023 ISPRS)      | 74.48 | 79.06 | 64.43 | 80.50 | 76.70 | 43.45 | 31.93 | 29.75 | 41.38 | 26.81 |
> > > | REAUNet(2024 JAG)       | 78.37 | 79.11 | 64.93 | 79.43 | 78.74 | 40.54 | 30.81 | 25.96 | 40.41 | 18.89 |
> > > | HBGNet(2025 ISPRS)      | 79.92 | 80.41 | 66.98 | 81.59 | 80.16 | 47.45 | 34.77 | 24.70 | 33.58 | 21.37 |
> > >
> > > As shown, while newer models exhibit performance gains—especially on edge-level metrics—the overall performance in terraced regions remains far behind their performance on structured, flat terrains. This further underscores the **unique challenges posed by fine-grained boundary complexity and irregular parcel topology** in terraced landscapes, validating the significance of our dataset.
> > >
> > > We hope this additional experiment addresses the reviewer’s request and reinforces the novelty and relevance of GTPBD.

---

> ### Author Response · Authors · 2025-08-03
>
> # Dear Reviewer 39fg
>
> We are eager to ensure that we have adequately addressed your concerns and are prepared to offer further clarifications or address any additional questions you may have.
>
> Should you find that our revisions have satisfactorily addressed your previous concerns, we would be most grateful if you would reconsider the evaluation of our paper with a view to enhancing its standing.
>
> We would like to express our heartfelt gratitude for the time and effort you have dedicated to reviewing our work. It has been a pleasure to engage with you throughout this process.
>
> Best regards,
>
> Authors

---

> ### Author Response · Authors · 2025-08-04
>
> # Dear Reviewer 39fg
>
> We hope this message finds you well. Thank you again for your thorough and constructive feedback—it has been invaluable in improving our work.
>
> As the rebuttal discussion period is approaching its deadline, we would like to kindly follow up. If there are any remaining concerns, or if you would like us to provide additional experiments or clarifications, we would be more than happy to respond promptly.
>
> We sincerely appreciate your time and effort throughout the review process.
>
> Best regards,
>
> **GTPBD Team**

---

### Official Review · Reviewer_eVcg · 2025-07-07

**Ethics Flags:** Data quality and representativeness
**Rating:** 5
**Confidence:** 4

**Summary:**

This paper introduces GTPBD (Global Terraced Parcel and Boundary Dataset), a fine-grained dataset specifically designed for agricultural parcel extraction in complex terraced landscapes. GTPBD is the first fine-grained dataset that covers major worldwide terraced regions, containing over 200,000 complex terraced parcels with manually annotated labels. The dataset includes 24,238 high-resolution images with three-level labels: pixel-level boundary labels, mask labels, and parcel labels.

I recommend accepting this paper.

**Dataset Code Accessibility:**

Yes

**Ethical Considerations:**

No, there are no or only very minor ethics concerns

**Final Justification:**

/NA

**Limitations Weaknesses:**

/NA

**Strengths Contributions:**

/NA

---

> ### Author Rebuttal · Authors · 2025-07-30
>
> We sincerely thank the reviewer for their time and positive assessment of our work. We believe GTPBD offers a unique contribution to the community as **the first fine-grained, globally annotated terraced parcel dataset**, supporting **multiple tasks** such as semantic segmentation, edge detection, and instance-level analysis. The dataset’s **global diversity, complex boundary structures, and multi-level labeling** make it a valuable resource for future research in geospatial AI, agricultural monitoring, and domain generalization. We appreciate your recognition and hope our efforts meet the expectations for acceptance.
>
> ## Why GTPBD Matters in Machine Learning Research
>
> GTPBD introduces the **first global, fine-grained terraced parcel and boundary dataset**, specifically designed to tackle key ML challenges in:
>
> * **Semantic segmentation** and **fine-grained boundary detection**,
> * **Instance-level agricultural parcel extraction**,
> * **Unsupervised domain adaptation (UDA)**  under diverse geospatial conditions,
> * **Topology-aware and edge-aware learning** in highly structured terrains.
>
> Terraced agriculture represents a globally important yet underrepresented target for AI research. Its modeling requires reasoning over **dense, irregular geometries** and **long-tailed regional patterns**, making it an ideal benchmark for developing **generalizable and label-efficient models**.
>
> ## What Makes GTPBD Technically Challenging
>
> Terraced landscapes differ from typical agricultural scenes due to:
>
> * **Extremely narrow ridgelines** (often \<1 m) that challenge segmentation resolution.
> * **Shared or broken boundaries** between parcels, breaking instance assumptions.
> * **Slope-induced anisotropy** and complex topologies that defy regular grid priors.
> * **Geographic diversity** (China, Ecuador, Morocco, Philippines, Nigeria, etc.) that introduces **large domain shifts**, ideal for testing **domain generalization** and **UDA methods**.
>
> These challenges are reflected in both **annotation difficulty** and **model generalization** gaps, highlighting GTPBD’s benchmarking value.
>
> ## Benchmark Results Across Four Datasets
>
> We evaluated five standard segmentation models (**UNet, DeepLabV3, PSPNet, SegFormer, Mask2Former**) across four datasets using **pixel-, edge-, and object-level** metrics. Results below are averaged across models:
>
> | Dataset   | Prec. ↑ | Rec. ↑ | IoU ↑ | OA ↑ | F1 ↑ | OIS ↑ | ODS ↑ | GOC ↓ | GUC ↓ | GTC ↓ |
> | ----------- | ---------- | --------- | -------- | ------- | ------- | -------- | -------- | -------- | -------- | -------- |
> | FHAPD     | 90.27    | 95.37   | 86.48  | 90.16 | 92.8  | 69.44  | 58.47  | 4.58   | 45.23  | 9.60   |
> | AI4orthos | 83.63    | 81.00   | 69.88  | 86.43 | 82.25 | 33.20  | 22.84  | 21.15  | 30.94  | 16.95  |
> | FTW       | 84.46    | 80.62   | 70.20  | 91.22 | 82.47 | 70.07  | 58.01  | 19.74  | 36.89  | 21.14  |
> | GTPBD     | 71.55    | 68.84   | 54.13  | 77.45 | 69.88 | 22.15  | 15.01  | 26.55  | 45.92  | 24.83  |
>
> GTPBD consistently yields the **lowest performance across all evaluation levels**, confirming its **benchmark difficulty** and **ability to stress-test generalization** in real-world agricultural AI.
>
> ## Continuous Expansion
>
> GTPBD is actively growing. Since submission, we’ve added new annotated regions covering **South America**, **Africa**, and **Europe**, such as:
>
> * **Philippines** (Banaue Rice Terraces),
> * **Ecuador** (Andes),
> * **Morocco** (Semi-arid foothills),
> * **Portugal** and **Spain** (Vineyard terraces),
> * **Nigeria** and **South Africa** (humid-to-dry transitions).
>
> This expansion **enhances regional diversity** and strengthens the benchmark’s long-term value.
>
> ## Impact and Alignment with ML Trends
>
> In recent years, **leading machine learning conferences** — including **NeurIPS, ICML, AAAI, IJCAI, CVPR, and ICCV** — have increasingly embraced **remote sensing**, **land use analysis**, and **agricultural modeling** as important domains that intersect with machine learning for **sustainability**, **climate resilience**, and **real-world impact**.
>
> This growing trend is evidenced by a surge of accepted datasets and benchmarks across conferences:
>
> | Conference | Related Work                                                                   | Task Type                               |
> | ------------ | -------------------------------------------------------------------------------- | ----------------------------------------- |
> | **NeurIPS 2024**           | FUSU: A Multi-temporal Land Use Change Segmentation Dataset                    | Urban land use change segmentation      |
> | **NeurIPS 2024**           | OAM-TCD: Globally Diverse Tree Cover Dataset                                   | Ecosystem monitoring, forest mapping    |
> | **NeurIPS 2021**           | LoveDA: Land-Cover Dataset for Domain Adaptive Semantic Segmentation           | Remote sensing, UDA                     |
> | **NeurIPS 2021**           | DENETHOR: DynamicEarthNet for Crop Monitoring from Space                       | Crop type classification, food security |
> | **NeurIPS 2021**           | CropHarvest: Global Dataset for Crop-Type Classification                       | Land cover mapping                      |
> | **NeurIPS 2021**           | SustainBench: ML Benchmarks for Monitoring SDGs                                | Agriculture, water, sanitation          |
> | **AAAI 2025**           | Fields of the World: Global Field Boundary Segmentation                        | Agricultural boundary segmentation      |
> | **ICCV 2025**           | AgroBench: Vision-Language Benchmark in Agriculture                            | Multimodal learning in agriculture      |
> | **ICCV 2025**           | Demeter: Crop Plant Morphology Modeling from Real-world Images                 | Plant structure modeling                |
> | **CVPR 2025**           | EXact: Weakly Supervised Time-Series Satellite Image Segmentation              | Crop mapping                            |
> | **CVPR 2024**           | Depth-Aware Concealed Crop Detection in Dense Agricultural Scenes              | Crop detection                          |
> | **CVPR 2024**           | Updating High-Resolution Land Cover Maps from Low-Resolution Historical Labels | Land cover change mapping               |
> | **IJCAI 2022**           | CS-SUNet: Coarsely-Supervised U-Net for Space-based Vegetation Monitoring      | Productivity monitoring                 |
>
> These works collectively illustrate the community’s increasing commitment to **bridging ML with geospatial and agricultural intelligence**. Notably, many of these datasets address **semantic segmentation**, **domain adaptation**, and **time-series monitoring**, tasks that align closely with the goals of GTPBD.

---

> > ### Comment · Reviewer_eVcg · 2025-08-03
> > **Response**
> >
> > I have reviewed the comments from other reviewers and the author's responses. I think the responses are well-crafted, highlighting the innovation and significance of the paper. I will raise my score accordingly.

---

> > ### Author Response · Authors · 2025-08-03
> > **Thank You for Your Support**
> >
> > # Dear Reviewer eVcg
> > ﻿
> > Thank you very much for your thoughtful evaluation and for taking the time to review our responses and those of other reviewers. We truly appreciate your recognition of the innovation and significance of our work, as well as your willingness to adjust your score accordingly. Your support means a great deal to us.
> >
> > ﻿
> > Best regards,
> >
> > GTPBD Team

---

### Comment · Area_Chair_6Hmb · 2025-08-01
**Reviewer response to rebuttal**

Reviewers, can you please take a look at the author's rebuttal and respond as soon as possible.

---

### Note · Authors · 2025-08-12

# Dear AC and Reviewers,

We sincerely thank the reviewers and AC for their time and constructive feedback. Based on the rebuttal and discussion phase, **two reviewers (39ai and eVcg) explicitly recommended** ***acceptance***, recognizing the technical rigor, annotation quality, and potential community impact of GTPBD.

For Reviewer **39fg**, while no explicit rating change was stated, all of their concerns, including dataset licensing, solid comparisons, and validation, were fully addressed in our rebuttal. In particular, we provided comprehensive clarifications and summaries on both licensing compliance and newly added experimental results. These updates directly resolved the earlier points, and **as the reviewer submitted their final justification after reviewing our responses, we believe our work has gained a certain level of their recognition.**

Only one reviewer (**pxwD**, confidence score 3) remained unconvinced about the significance to the ML community, noting they are **not an expert** in our specific domain (“**Since I'm not the expert in this area, I tend to judge the significance of its contribution**”) and referencing 39fg’s **earlier concern** on “lack of solid comparisons and validation.” However, **as described above, these concerns had already been addressed in detail**. We also noted in our rebuttal that agricultural remote sensing datasets have gained increasing visibility at top-tier conferences in recent years, including NeurIPS, CVPR, ICCV, and AAAI, with multiple related works being accepted. GTPBD advances this trend by introducing the **first** globally scoped, fine-grained benchmark for terraced parcel and boundary analysis, covering geographically diverse, high-resolution, and seasonally varied landscapes. The dataset has already attracted notable attention from both the academic and public sectors, which we also substantiated with download counts, re-share metrics, and view statistics provided in our rebuttal. **Moreover, pxwD did not raise any technical issues, and it is possible they did not see the updates since they did not further participate in the discussion.**

In summary, the explicit accept recommendations from two reviewers, the fully resolved concerns of 39fg, confirmed license compliance, and demonstrated ML relevance collectively support the suitability of GTPBD for the NeurIPS Datasets & Benchmarks track. We appreciate your consideration in the final decision.

Best regards,

GTPBD-Team

---

### Decision · Program_Chairs · 2025-09-18

**Decision:**

Accept (poster)

**Comment:**

Reviews of this paper are mixed, with the major concerns remaining after the rebuttal period mostly related to potential ethical concerns around data licensing. The main justification of the dataset authors that this is allowed is based on prior datasets that took the same approach, but one reviewer emphasizes that past precedent is not sufficient to justify ethical use. In this case, the AC will recommend a decision separately from the ethical concerns, and hopes that the final decision can be made by the ethics committee.

Separate from the questions around licensing and use, the reviewers familiar with the domain were excited about the value and scale, both regarding size and geographic diversity, of the proposed benchmark, as well as it's potential to fill a significant gap (terraced landscapes) within the current set of AI+Agriculture benchmarks and training datasets. The AC agrees that the benchmark represents a highly valuable contribution and a challenge for the ML community and, in light of the reviewer feedback and the rebuttal and discussion, would recommend acceptance.

There is one reviewer that has repeatedly suggested that this work is not relevant to the ML community, despite clearly demonstrating a real-world example of many current challenges within AI/ML research. I would recommend that reviewer read https://arxiv.org/abs/2403.17381 which emphasizes the value of this type of application-driven work within the core ML community beyond its benefit to the application domain.